# Room-temperature intrinsic ferromagnetism in epitaxial CrTe$_2$ ultrathin films

Xiaoqian Zhang[1,2,12], Qiangsheng Lu[2,12], Wenqing Liu[1,3,12], Wei Niu[4], Jiabao Sun [3], Jacob Cook[2], Mitchel Vaninger[2], Paul F. Miceli[2], David J. Singh [2,5], Shang-Wei Lian[6], Tay-Rong Chang [6,7], Xiaoqing He[8,9], Jun Du[10], Liang He [1✉], Rong Zhang [1✉], Guang Bian [2✉] & Yongbing Xu [1,11✉]

While the discovery of two-dimensional (2D) magnets opens the door for fundamental physics and next-generation spintronics, it is technically challenging to achieve the room-temperature ferromagnetic (FM) order in a way compatible with potential device applications. Here, we report the growth and properties of single- and few-layer CrTe$_2$, a van der Waals (vdW) material, on bilayer graphene by molecular beam epitaxy (MBE). Intrinsic ferromagnetism with a Curie temperature ($T_C$) up to 300 K, an atomic magnetic moment of ~0.21 $\mu_B$/Cr and perpendicular magnetic anisotropy (PMA) constant ($K_u$) of 4.89 × 10$^5$ erg/cm$^3$ at room temperature in these few-monolayer films have been unambiguously evidenced by superconducting quantum interference device and X-ray magnetic circular dichroism. This intrinsic ferromagnetism has also been identified by the splitting of majority and minority band dispersions with ~0.2 eV at $\Gamma$ point using angle-resolved photoemission spectroscopy. The FM order is preserved with the film thickness down to a monolayer ($T_C$ ~ 200 K), benefiting from the strong PMA and weak interlayer coupling. The successful MBE growth of 2D FM CrTe$_2$ films with room-temperature ferromagnetism opens a new avenue for developing large-scale 2D magnet-based spintronics devices.

[1] Jiangsu Provincial Key Laboratory of Advanced Photonic and Electronic Materials, School of Electronic Science and Engineering, Nanjing University, Nanjing, China. [2] Department of Physics and Astronomy, University of Missouri, Columbia, MO, USA. [3] Department of Electronic Engineering, Royal Holloway University of London, Egham, Surrey, UK. [4] New Energy Technology Engineering Laboratory of Jiangsu Provence & School of Science, Nanjing University of Posts and Telecommunications, Nanjing, China. [5] Department of Chemistry, University of Missouri, Columbia, MO, USA. [6] Department of Physics, National Cheng Kung University, Tainan, Taiwan. [7] Center for Quantum Frontiers of Research and Technology (QFort), Tainan, Taiwan. [8] Electron Microscopy Core Facility, University of Missouri, Columbia, MO, USA. [9] Department of Mechanical and Aerospace Engineering, University of Missouri, Columbia, MO, USA. [10] National Laboratory of Solid State Microstructures and Department of Physics, Nanjing University, Nanjing, China. [11] York-Nanjing Joint Centre (YNJC) for Spintronics and Nano Engineering, Department of Electronic Engineering, The University of York, York, UK. [12] These authors contributed equally: Xiaoqian Zhang, Qiangsheng Lu, Wenqing Liu. ✉email: heliang@nju.edu.cn; rzhang@nju.edu.cn; biang@missouri.edu; ybxu@nju.edu.cn

Two-dimensional (2D) layered magnets exhibit novel phases of quantum matter with abrupt transition in the magnon density of states in atomically thin layers. In a three-dimensional (3D) system, the magnon density of states are consecutive and chiefly determined by exchange interactions. Therefore, a magnetic phase transition could occur at a finite temperature. By contrast, the long-range magnetic order in 2D systems is fragile against thermal fluctuations according to the Mermin–Wagner theorem[1,2]. The magneto-anisotropy in 2D ferromagnets opens up a large spin-wave excitation gap, quenches thermal fluctuations[3–9], and thus stabilizes the long-range magnetic order in 2D regime. In contrast to defect or dopant induced magnetism, the ferromagnetism occurring in a stoichiometric compound is defined as intrinsic ferromagnetism[10].

While the presence of 2D crystals with intrinsic magnetism has been well established, the intrinsic ferromagnetic (FM) order in the discovered magnetic van der Waals (vdW) materials is generally fragile with a low Curie temperature ($T_C$). It mainly results from the enhanced spin fluctuation in reduced dimensions or the relatively weak exchange interactions. Note that the interlayer bonding strength in vdW compounds is 2–3 orders of magnitude weaker than that of traditional 3D materials[4], which leads to a low $T_C$ in the bulk form already. It motivates research efforts to enhance the robustness of 2D FM order. The first route is doping a FM host with specific elements, which normally results in a limited increase of $T_C$ but unavoidable clusters and/or disorders from dopants[11,12]. The second one is constructing heterostructures with FM (or ferrimagnetic) metals (or insulators), in which the FM order can be enhanced by proximity effects[13,14]. For instance, the (Fe$_3$GeTe$_2$/MnTe)$_3$ superlattices possess an enhanced coercive field as a result of the proximity effect[12]. However, the penetration depth of proximity effect is usually very small (<5 nm), hindering an effective manipulation of magnetic order. The third method is doping 2D magnets with electrons via electrolyte gating, and thereby modulating the $T_C$ of ferromagnetism. For example, the $T_C$ of an atomically thin Fe$_3$GeTe$_2$ flake is successfully raised to even room temperature[15]. Nevertheless, particular device geometry and gating are required by this means. Apart from the issues mentioned above, most of the 2D magnetic materials reported so far are thin flakes exfoliated from bulk with typical size of several micrometers, which greatly limits the practical applications of those 2D magnets in spintronics. Therefore, there is a pressing need for the realization of stoichiometric 2D materials with intrinsic robust ferromagnetism (e.g., high $T_C$ and strong perpendicular anisotropy) and, importantly, compatibility with large-scale solid-state device applications.

Molecular beam epitaxy (MBE) growth is significant as it provides the opportunity to obtain nominally stoichiometric single-crystalline films, explore the role of physical dimensionality as well as fabricate heterostructures and superlattices in a way compatible with conventional microelectronics techniques. One remarkable work is the strong FM order in ML VSe$_2$ epitaxial film with in-plane easy axis and a large magnetic moment (~15 $\mu_B$/V) persisting to even above room temperature, as characterized by magneto-optical Kerr effect (MOKE) and vibrating sample magnetometry (VSM)[16]. However, according to the theoretical calculations, the magnetic moment of ML VSe$_2$ mostly comes from V ions with an atomic value of ~0.6 $\mu_B$[17], which is completely contradictory to the experimentally observed large magnetic moment[16], raising doubts about this presumed FM phase. Most recently, Wong et al. has provided the evidence of spin frustration with absence of a long-range magnetic order in ML VSe$_2$ films from complementary temperature- and field-dependent susceptibility measurements[18], in stark contrast to the previous study. Moreover, the electronic structure and X-ray magnetic circular dichroism (XMCD) measurements of ML VSe$_2$

conducted by Feng et al. reveal no signatures of FM order[19]. These studies suggest that the existence of 2D FM order in VSe$_2$ remains to be further confirmed. Therefore, layer-controlled growth of stoichiometric large-scale 2D FM films with strong perpendicular magnetic anisotropy (PMA) and direct proof for the intrinsic ferromagnetism by unambiguous techniques would be mandatory. Notably, an above-room-temperature $T_C$ has been reported in 1T-CrTe$_2$ in its bulk form[20]. Very recently, above-room-temperature ferromagnetism has been observed in the exfoliated thin flakes of CrTe$_2$ (10 nm, or ~17 ML)[21,22]. Their properties were found to be rather similar to that of the bulk with in-plane magnetic anisotropy, but with enhanced coercivity compared with its bulk counterpart. However, the magnetic response (e.g., $T_C$ and PMA) of CrTe$_2$ epitaxial thin films with thickness down to ML limit has not been explored so far.

In this work, we succeed in synthesizing mono- and few-layer CrTe$_2$ films by MBE and observed intrinsic long-range 2D ferromagnetism. The robust ferromagnetism and strong PMA of CrTe$_2$ films persist up to 300 K, as evidenced by both superconducting quantum interference device (SQUID) and XMCD characterizations. In addition, the splitting of the majority and minority bands (~0.2 eV at Γ point) with distinct photon-energy responses has been observed by in-situ angle-resolved photoemission spectroscopy (ARPES) measurements, suggesting the magnetic band structure of CrTe$_2$ with spin-splitting. Furthermore, the CrTe$_2$ thin films retain a robust ferromagnetism with high $T_C$ down to a ML, indicating a weak dimensionality effect. These results establish CrTe$_2$ ultrathin films as a promising 2D ferromagnet for exotic low-dimensional spintronics applications.

## Results

CrTe$_2$ is a layered trigonal crystal structure with a unit cell of a hexagonal Cr layer sandwiched between Te layers, as schematically illustrated in Fig. 1a. In our experiment, a bilayer graphene on SiC substrate was used to support a layer-by-layer growth of CrTe$_2$ films. The optical image of a single-crystal CrTe$_2$ film with large size (~4 mm × 5 mm) is shown in the inset of Fig. 1b. The microscopic topography taken from the surface of a few-layer CrTe$_2$ film by in-situ scanning tunneling microscopy (STM) shows atomically flat terrace (Fig. 1b). Figure 1c exhibits the step height between adjacent layers with a uniform value of 6.14 Å, which is consistent with the thickness of the unit cell of CrTe$_2$ crystal in 1T phase. One of the atomic resolution image taken by STM on the same sample is presented in Fig. 1e, showing the hexagonal lattice structure. The lattice constant obtained from the line profile in Fig. 1f is 3.81 Å, which is very close to the corresponding bulk CrTe$_2$ lattice parameter (3.79 Å)[20]. STM measurements carried out on several CrTe$_2$ thin films with different thicknesses (mono- to few-layer) show similar terraces, indicating the layer-by-layer growth mode and homogeneously well-structured thin films (see Supplementary Fig. 1).

There are various stable stoichiometries reported for chromium chalcogenides [e.g., CrTe[23,24], Cr$_2$Te$_3$[25,26], and Cr$_5$Te$_8$[27,28]] depending on the Cr vacancies that occur in intercalation. However, none of them belongs to layered compounds with interlayer vdW gap, except for CrTe$_2$. The layered surface morphology with a uniform step height characterized by STM suggests that the films are in a single phase with vdW gap. The atomic-resolution high-angle annular dark-field scanning transmission electron microscopy (HAADF-STEM) images show the $\sqrt{3}a \times a$ arrangement, revealing that CrTe$_2$ thin films correspond to the 1T phase with an octahedral ($O_h$) symmetry (see Supplementary Fig. 2). Both TEM and STM characterizations manifest the epitaxial nature and crystallographic orientation of as-grown CrTe$_2$ films. A typical X-ray diffraction (XRD) $2\theta$-$\omega$ scan was

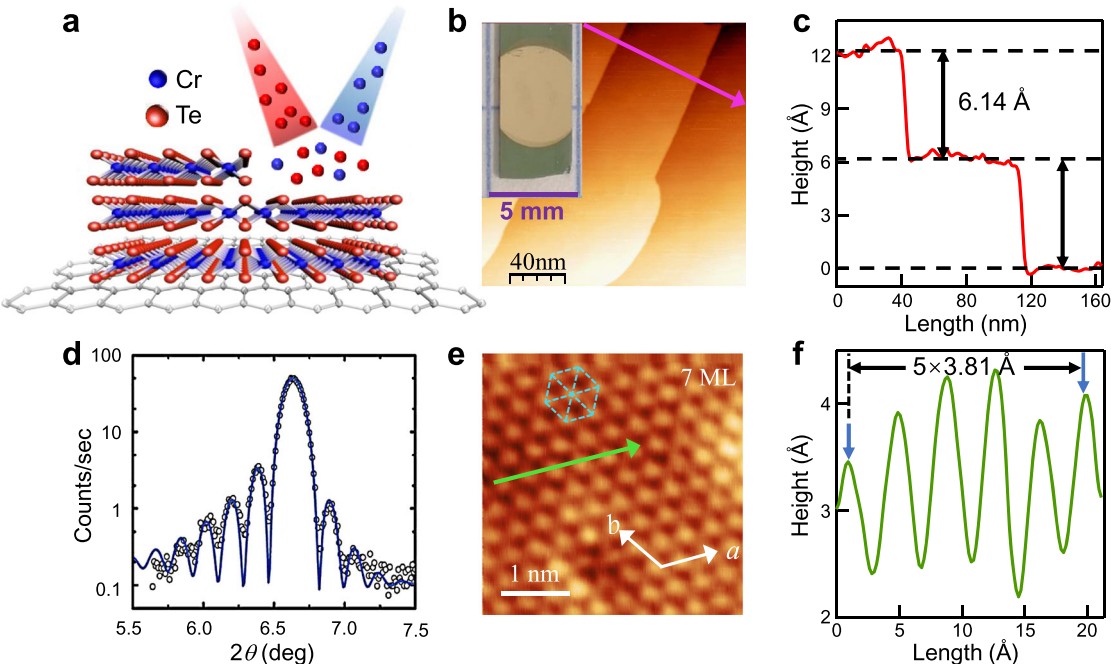

**Fig. 1 Crystal structure and STM characterizations of epitaxially grown CrTe$_2$ thin films. a** Schematic illustration of MBE growth process of CrTe$_2$ films on graphene. **b** The STM topology image (200 × 200 nm$^2$) of a 7 ML CrTe$_2$ fabricated on graphene/SiC. U = +1 V, $I_t$ = 200 pA. Inset on the left is an optical image. **c** The line-scan profile taken along the pink line in (**b**), with an average step height of ~6.14 Å. **d** XRD spectrum showing Laue fringes around the (001) CrTe$_2$ reflections. The solid fitting curve indicates the thickness of 39 layers, the roughness of 2 layers, and the lattice constant c = 6.13 Å. **e** Atomically resolved STM image (4 × 4 nm$^2$) with a hexagonal structure. U = −1.5 mV, $I_t$ = −440 pA. **f** The line-scan along the green arrow in (**e**), showing a lattice periodicity of ~3.81 Å.

employed to further identify the crystal structure (Fig. 1d). The diffraction pattern with perpendicular constant $c = 6.13$ Å is matched to the (001) crystal planes of 1T-type hexagonal structure explored experimentally ($a = 3.79$ Å, $c = 6.10$ Å)[20], rather than those of the 2H phase ($a = 3.49$ Å, $c = 13.64$ Å)[29]. We note that the magnetic exchange coupling is sensitive to the lattice parameters. For example, bulk 1T-CrSe$_2$ with lattice constants of $a = 3.39$ Å and $c = 5.92$ Å shows an antiferromagnetic (AFM) order[30], in contrast to the FM phase in CrTe$_2$. With STM, TEM, and XRD characterizations, the formation of CrTe$_2$ films with 1T phase and their single-crystalline nature has been confirmed. The reflectivity curves show Laue fringes, attesting to the structural coherence of the film. The chemical states and band structure of the as-synthesized samples were determined by X-ray absorption spectroscopy (XAS) and ARPES as included in the following part, respectively, which further identify the metallic 1T phase in these few-layer CrTe$_2$ films.

Magnetic properties of CrTe$_2$ thin films with both in-plane and out-of-plane configurations were examined by SQUID, as shown in Fig. 2. The temperature dependent magnetization (**M–T**) curves of CrTe$_2$ thin films with different thicknesses under an out-of-plane magnetic field of 1000 Oe were measured, as shown in Fig. 2a. It shows a general trend of decreasing with the increase of temperature, demonstrating a FM nature. It indicates that the $T_C$ is close to the room temperature with the specific values depending on the thickness. The magnetization of 7 ML CrTe$_2$ film is still observable at 300 K, indicating the FM order at room temperature. The magnetization curve exhibits a long "tail" near $T_C$, which is commonly observed in ferromagnets[4,11,31]. It can be explained by a positive-feedback mean-field modification of the classical Brillouin magnetization theory[32].

The magnetization-magnetic field (**M–H**) hysteresis loops acquired from the 7 ML CrTe$_2$ film at different temperatures are included in Fig. 2b and c. The sharp distinction between out-of-

plane (Fig. 2b) and in-plane (Fig. 2c) **M–H** loops demonstrates a strong out-of-plane anisotropy of the magnetization with a large PMA constant ($K_u = \frac{H_k M_s}{2}$) of $5.63 \times 10^6$ erg/cm$^3$ at 20 K. The $K_u$ in CrTe$_2$ thin films is comparable to the typical PMA systems such as Co/Pd and Co/Pt (see Supplementary Table 1)[33–36], which is vital for obtaining 2D FM order and is also considerably desirable for vdW heterostructures-based spintronics. The film also exhibits rather large coercivities (e.g., ~1000 Oe at 20 K), indicative of a hard magnetic phase. Well-defined hysteresis loops are observed at elevated temperatures up to 300 K (Fig. 2d) with the easy axis along the out-of-plane direction and hard axis along the in-plane one. The existence of PMA in the ultrathin 7 ML film is confirmed, supporting the FM order at room temperature. The in-plane magnetic hysteresis loops, similar to those reported in the FM vdW Cr$_2$Ge$_2$Te$_6$ thin films[37] and typical PMA systems such as Mn$_{2.5}$Ga[38] and Co/Pt[39], can be attributed to the shape anisotropy favoring in-plane easy axis for thin films[40,41]. Control experiments on the field dependent magnetization of SiC/graphene substrate show a typical diamagnetic behavior (see Supplementary Fig. 4). Therefore, the possibility of magnetic contribution from magnetic impurities in the substrate can be ruled out. In order to clarify the thickness dependence of the magnetic properties, we have measured the field dependent magnetization curves of 3 ML and 5 ML CrTe$_2$ thin films under out-of-plane and in-plane configuration (Supplementary Fig. 5). The squarish FM hysteresis loops in the out-of-plane magnetic field suggest the robust FM order with the easy axis perpendicular to the thin films, which is essential for the applications of FM devices. Compared with other 2D magnets from literature[3,4,42,43], the CrTe$_2$ films perform a relatively high $T_C$ (above room temperature) and strong magnetic anisotropy with a few atomic layers. Notably, a large $K_u$ ($4.89 \times 10^5$ erg/cm$^3$) is maintained at 300 K, comparable to the value of bulk CrGeTe$_3$ at 1.8 K ($4.7 \times 10^5$ erg/cm$^3$)[44]. The strong PMA in CrTe$_2$ few-layer films is

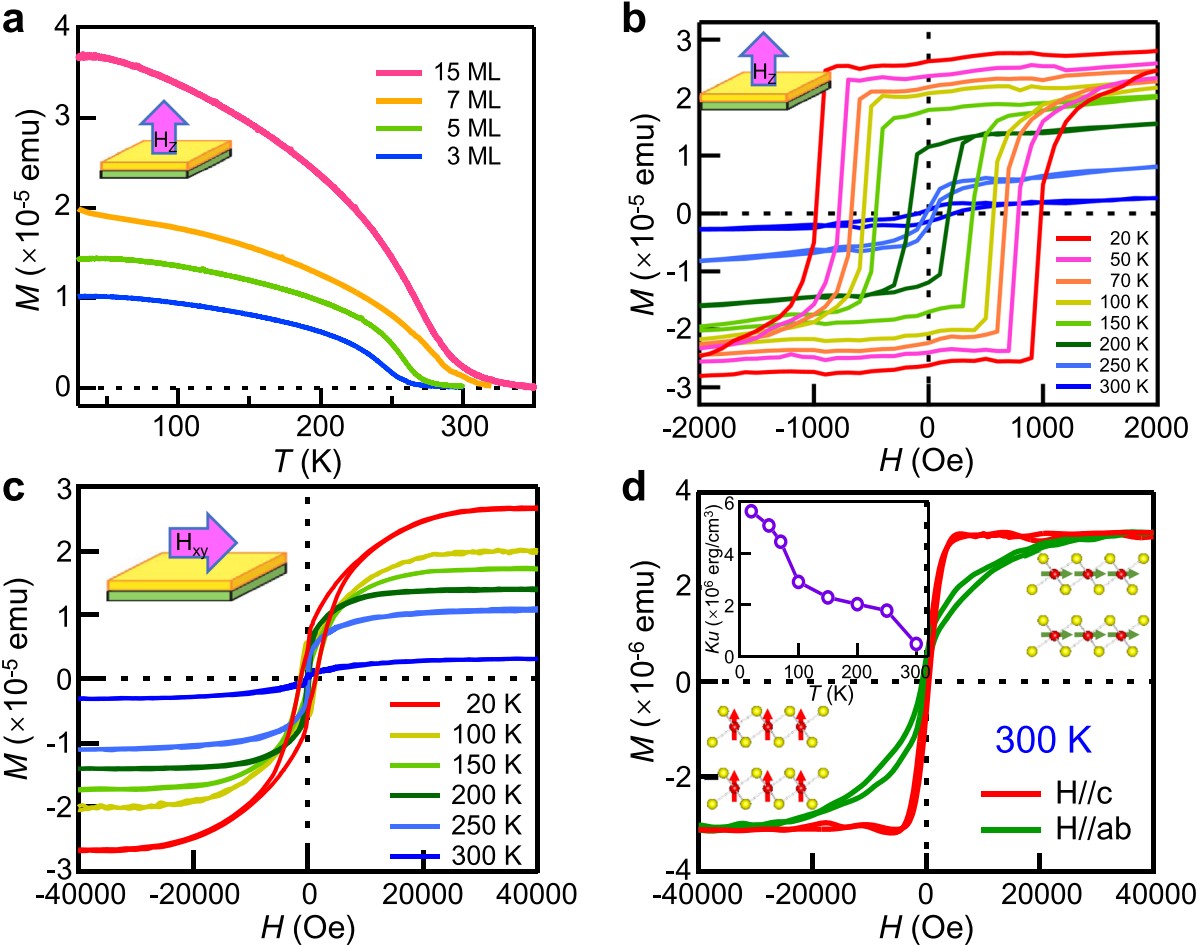

**Fig. 2 SQUID measurements of the CrTe₂ films. a** Temperature dependent magnetization curves of the films with various thicknesses under field-cooled mode. The magnetic field is applied along the out-of-plane direction with a magnitude of 1000 Oe. The high $T_C$ is preserved with thickness decreasing to 3 ML. **b, c** Magnetic hysteresis loops of 7 ML CrTe₂ at different temperatures with external fields along the perpendicular (**b**) and parallel orientation (**c**) with respect to sample plane, indicating a strong out-of-plane magnetic anisotropy. **d** Enlarged hysteresis loops of 7 ML CrTe₂ at 300 K, where the intrinsic ferromagnetism and PMA still maintains. Top inset: temperature dependence of $K_u$ for 7 ML CrTe₂, where the $K_u$ is preserved at 300 K, despite the lower intensity with the increase of temperature.

different from the in-plane magnetic anisotropy observed in bulk CrTe₂[20] and exfoliated flakes (thicker than 10 nm)[21]. Here, the thickness dependent magnetic anisotropy suggests that the reduced symmetry at the interface plays an important role in determining the PMA in CrTe₂ thin films[45]. As the magnetic film thickness approaches a few nm, the interfacial magnetism and inversion symmetry breaking give rise to the PMA[46]. This is a consequence of magneto-crystalline anisotropy from spin–orbit interactions, which apparently have a stronger effect in the more anisotropic film limit[33,41,45]. In addition, based on density functional theory (DFT) calculations, it has been found that the FM Cr–Cr intrasublattice exchange interactions dominate in CrTe₂ thin films, and the total energy minima is at perpendicular direction[47]. In general, the magnetic moments of CrTe₂ thin films are aligned in the perpendicular direction, due to the magneto-crystalline anisotropy and the anisotropy of exchange interactions.

To examine the local electronic character and magnetic ground states of CrTe₂ films, XAS and XMCD measurements at the Cr $L_{2,3}$ absorption edges were performed, as schematically shown in Fig. 3a. This element-specific magnetic characterization technique can also exclude any possible magnetic impurities. The XAS spectra of Cr present multiplet structures around photon energies of 575 eV and 584 eV (Fig. 3b), which stem from the excitations

from Cr $2p_{3/2}$ and Cr $2p_{1/2}$ core levels, respectively. A small peak (~2 eV away from $L_3$ peak of Cr) marked with the black arrow comes from Te $5d_{5/2}$ core level, which slightly overlaps with the peak of Cr $2p_{3/2}$ with almost no magnetic contribution[48]. A small peak at the higher energy side (marked with orange arrow) of the main feature in Fig. 3b is related to the distribution of atomic multiplet. Due to the $O_h$ coordination, the $3d$ orbitals of Cr split into $e_g$ and $t_{2g}$ states with energy separation of nominal 10 $Dq$. The $t_{2g}$ states are lower in energy than the $e_g$ states. In this case, the $Cr^{3+}$ ($d^3$) configuration with half-filled $t_{2g}$ states causes the reduction of free energy[49], which is in good agreement with the reported theoretical value of magnetic moment, 3 $\mu_B$/Cr atom[50]. The observed XAS spectral line shape is in line with that of spinel $Cu(Cr,Ti)_2Se_4$ polycrystals with trivalent Cr cations on $O_h$ sites[51], further providing a spectroscopic fingerprint of 1T-type CrTe₂ with predominately $Cr^{3+}$ cations. In this case, approximately three electrons are removed from the Cr atoms, and distributed over the Te.

The Cr $L_{2,3}$ XMCD spectra in the bottom panel of Fig. 3b highlight the emergence of intrinsic ferromagnetism from Cr atoms. XMCD and XAS measurements were repeated at elevated temperatures, and the dichroism of 7 ML thin film at Cr $L_3$ edge is evident up to 300 K. The characteristic peaks in the spectra remain at the same energy as the temperature rises, despite the

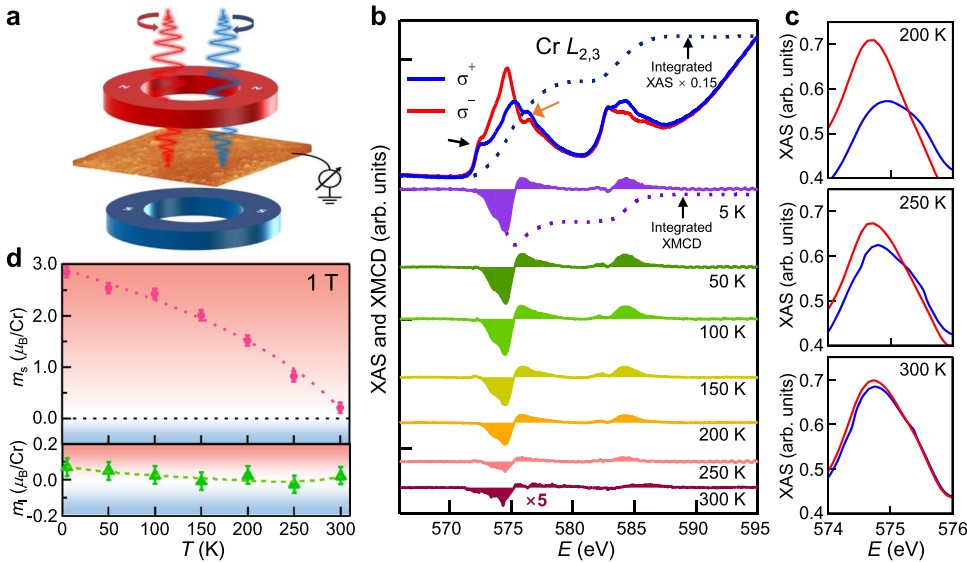

**Fig. 3 XAS and XMCD characterization of 7 ML CrTe₂ films. a** Schematic geometry of XMCD experimental setup. **b** Typical pairs of XAS and XMCD spectra of 7 ML CrTe₂ from 5 K to 300 K and the integrals at 5 K, where the dichroism at Cr $L_3$ edge can be traced to 300 K (spectra at different temperatures are offset for clarify). **c** The partially enlarged XAS of Cr $L_3$ edge at 200 K, 250 K, and 300 K. **d** $m_s$ and $m_l$ versus temperature derived from (**b**) using sum rules. The error bars reflect the uncertainties in the background estimation for the XMCD sum rules analysis.

attenuation of intensity. For greater clarity, partial enlarged left- and right-circularly polarized XAS of Cr $L_3$ edge at 200 K, 250 K, and 300 K are exhibited in Fig. 3c. There is an obvious difference between the XAS under distinct X-ray helicity even at 300 K, directly confirm the intrinsic FM order coming from the $Cr^{3+}$ cations in the CrTe₂ films. The XMCD spectra have been analyzed in terms of element-specific magnetic moments according to the sum rules[52,53]. The spin moment ($m_s$) and orbital moment ($m_l$) can be obtained by sum rule:

$$m_s = -n_h \frac{6\int_{L_3}(\sigma^+ - \sigma^-)dE - 4\int_{L_{2,3}}(\sigma^+ - \sigma^-)dE}{\int_{L_{2,3}}(\sigma^+ + \sigma^-)dE} \times SC - \langle T_z \rangle$$

(1)

$$m_l = -\frac{4}{3} n_h \frac{\int_{L_{2,3}}(\sigma^+ - \sigma^-)dE}{\int_{L_{2,3}}(\sigma^+ + \sigma^-)dE}$$

(2)

where $n_h$, SC and $\langle T_z \rangle$ are the number of $d$ holes, spin correction factor (estimated to be $2.0 \pm 0.2$ for Cr)[13,54] and the averaged magnetic dipole term, respectively. Based on the trivalent Cr, we assume $n_h = 7$. The magnetic dipole term, $\langle T_z \rangle$ can be neglected due to its rather small contribution (<5%) in the Cr $t_{2g}^3$ configuration. An arctangent step-like function was employed in the fitting of the threshold of XAS spectra in order to exclude the nonmagnetic contribution[55,56].

The calculated $m_s$ and $m_l$ from 5 to 300 K are summarized in Fig. 3d. The derived $m_s$ demonstrates a Curie-like behavior. A remarkably large value of $m_s$ ($2.85 \pm 0.10 \, \mu_B/Cr$) is found at 5 K. The $m_s$ retains a sizable value of $0.82 \pm 0.10 \, \mu_B/Cr$ at 250 K and drops to $0.21 \pm 0.05 \, \mu_B/atom$ at 300 K, confirming a FM phase transition near this temperature. On the other hand, $m_l$ is relatively small of around $0.08 \pm 0.05 \, \mu_B/atom$, consistent with a half-filled $t_{2g}$ level in $O_h$ crystal field of 1T-CrTe₂. The $m_l$ plays an important role in the magneto-crystalline anisotropy and the perpendicular orientation of the moments that underlies the FM order in this 2D system. The atomic magnetic moment of CrTe₂ is determined to be ~3 $\mu_B/atom$. The observed FM behavior cannot be attributed to the Cr clusters, since bulk Cr is AFM and therefore would give a zero XMCD intensity.

The magnetic response of 1 ML CrTe₂ film is worth exploring. It is difficult to detect magnetization in such ultrathin films by SQUID, since the magnetic signal of 1 ML CrTe₂ is too weak compared with an overwhelmingly larger background signal from the substrate and beyond the resolution of SQUID. Therefore, we did element-specific XMCD characterization of 1 ML CrTe₂ film (Fig. 4a). There is a noticeable difference in the XAS spectra between left- and right-handed circularly polarized setups (Fig. 4b). Although the dichroism is small compared with 7 ML sample, the clear XMCD signals appear near the absorption peaks. It suggests that the intrinsic ferromagnetism of 1 ML CrTe₂ film originates from the spin polarization of Cr $3d$ electrons. Accurate calculation of the magnetic moment remains a challenge since the contribution of Te capping layer to the XAS spectra is so large for 1 ML sample. The XMCD percentage increases with the reduced temperature (Fig. 4c), in line with a typical FM behavior. The nonzero XMCD percentage persists when temperature approaches 200 K and disappears above 250 K, indicating that 1 ML CrTe₂ has a $T_C$ of ~200 K. The $T_C$ has been obtained by using a critical power-law function $\alpha(1 - T/T_C)^\beta$ to fit **M-T** curves without the inclusion of the paramagnetic tail[42]. In order to investigate the dimensionality effect of the ferromagnetism in CrTe₂ stemming from thermal fluctuation, we plot the thickness dependent $T_C$ obtained from XMCD and SQUID measurements in Fig. 4d. The $T_C$ of CrTe₂ decreases mildly as the film thickness is reduced, in contrast to the other known 2D magnets such as Cr₂Ge₂Te₆[4] and Fe₃GeTe₂[15] (Supplementary Fig. 10). The high $T_C$ in the 2D limit demonstrates the robustness of ferromagnetism in the epitaxial CrTe₂ thin films.

The electronic band structure of CrTe₂ thin films has been mapped by ARPES with two different photon energies of 21.2 eV and 40.8 eV at 107 K. The band dispersions of 7 ML CrTe₂ measured at $h\nu = 21.2$ eV along high symmetry crystallographic direction M-Γ-K in the surface Brillouin zone are shown in Fig. 5a. Near the Γ point, the main features include two hole-like valence bands aligned close to the Fermi level, which shares identity with the typical features of 1T-ZrTe₂[57]. Near the M point, there are two electron pockets with bottom locating at −1.2 eV and −1.8 eV, respectively. The Fermi surface map shows two circular pockets centered at Γ point surrounded by six triangular

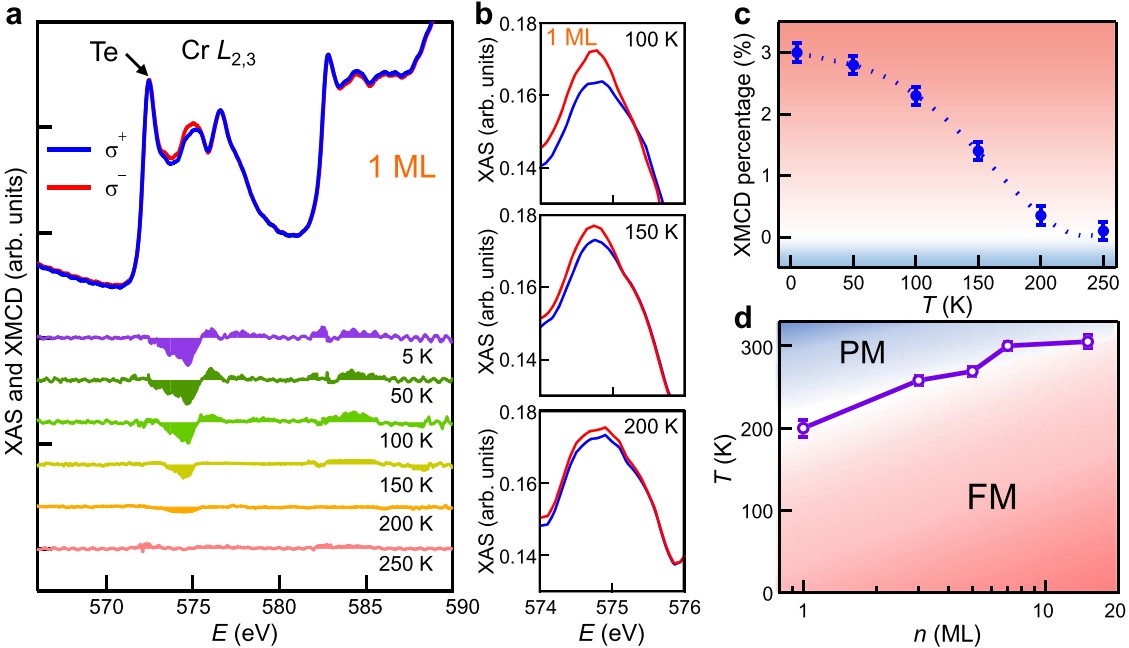

**Fig. 4 XAS and XMCD characterization of CrTe$_2$ films with thickness of monolayer. a** Typical pairs of XAS and XMCD spectra of 1 ML CrTe$_2$ thin film at various temperatures, where the dichroism at Cr $L_3$ edge is observable up to 200 K. **b** The partially enlarged XAS spectra near the Cr $L_3$ edge, where the difference between left- and right-circularly polarized XAS is evident. **c** XMCD percentage as a function of temperature derived from **a**. The error bars indicate the uncertainties in the background estimation for the XMCD percentage calculation. **d** Compiled thickness–temperature phase diagram with the $T_C$ obtained from XMCD and SQUID measurements. The error bars are the uncertainties in determining the $T_C$.

pockets at K points. Below the Fermi level, the pockets around K points begin to merge with the expanded pockets at Γ point (Supplementary Fig. 11). The well-defined band structure indicates the high structural quality of the MBE-fabricated films.

The origin of the band dispersions has been investigated by first-principle DFT calculations based on CrTe$_2$ slab[18]. The mean free path of photoelectrons excited by photons of 21.2 eV and 40.8 eV is between 0.5 and 1 nm. Therefore, to compare with the experimental spectra, we simulated the band structure with a surface weight of each Bloch wavefunction. The higher intensity in the image means greater weight of wavefunction near the slab surface. Figure 4b shows the calculated spin-polarized band structure, with the majority and minority spin bands plotted in blue and red, respectively. Both magnetization and spin–orbit coupling (SOC) are taken into account in the calculation, and the magnetic moments are set along out-of-plane direction. According to the orbital and surface projection analysis of the band structure, the metallicity is a consequence of the hybridization of Te-5$p$ and Cr-3$d$ orbitals crossing the Fermi level at the center of the Brillouin zone (see Supplementary Fig. 13), which is confirmed by the calculated density of states (see Supplementary Fig. 14). The hybridization of Te and Cr states was also verified in previous DFT calculations[20]. There is an overall agreement between the experimental (Fig. 5a) and calculated band dispersions (Fig. 5b), except for the absence of two hole pockets from minority band near Γ point.

To compare experiment and theory in greater detail, the dispersion of hole pockets detected by different photon energies is plotted in Fig. 5c and d. Note that the two hole pockets near the Fermi level in Fig. 5c are mainly from the majority bands. Interestingly, the minority hole pocket shows up in the spectrum taken at $hv = 40.8$ eV (Fig. 5d) while the majority ones disappear. It suggests the emission from the minority spin pockets was suppressed in the measurement at $hv = 21.2$ eV as a consequence of matrix element effect[58]. The band dispersion can be traced by fitting the peak position in the momentum distribution curves

(MDC), as marked by blue and red dashed lines in Fig. 5c and d, respectively. Combining the band structure near Fermi energy ($E_F$) taken by He Iα and He IIα photons together, as shown in Fig. 5e, the electronic structure is clearly metallic in both the majority and minority spin channels, and agrees well with DFT calculations. Relatively small renormalizations are needed to match with ARPES results, indicating moderate-to-weak correlations. The experimental band structure of CrTe$_2$ is in sharp contrast with the band structure calculated without the inclusion of spin polarization (Supplementary Fig. 15), where hole pockets near $E_F$ are degenerate at Γ point as in the cases of VTe$_2$[59] and VSe$_2$[18,19]. There are no exchange splitting of band dispersion in MBE grown VSe$_2$ films, indicating the absence of ferromagnetism[18,19]. By contrast, the splitting of majority and minority bands (~0.2 eV at Γ point) in CrTe$_2$ films corroborates the FM ground state, which highlights the unique interplay of ferromagnetism and electronic structure in CrTe$_2$. In addition, the calculated magnetic moment of Cr is 2.89 $\mu_B$/atom, in good agreement with the SQUID and XMCD measurements. For a comparison with the ARPES spectra, we also calculated the electronic band structure of 2H-CrTe$_2$, which are different from those observed in the ARPES spectra and the calculated 1T-CrTe$_2$ band structure in the low-energy bands (see Supplementary Fig. 17). Another significant difference between 1T and 2H phase is that the 1T-CrTe$_2$ exhibits a FM ground state along c-axis, while the 2H-CrTe$_2$ is PM as a result of the fully occupied $d_{z^2}$ orbital of tetravalent Cr. The observed FM band structure and FM properties corroborate the 1T phase of the epitaxial CrTe$_2$ films.

We have further studied the thickness dependence of hole pocket features. The evolution of the band structure for the films with a thickness ranging from 1 ML to 15 ML is shown in Fig. 5f. For the 1 ML film, there are two parabolic bands with a maximum above and below the Fermi level, respectively. When film thickness increases to 2 ML, one of the parabolic band overlaps with another one near the Fermi level, sharing similar feature with the case of few-layer ZrTe$_2$[57] and HfTe$_2$[60]. With further increasing

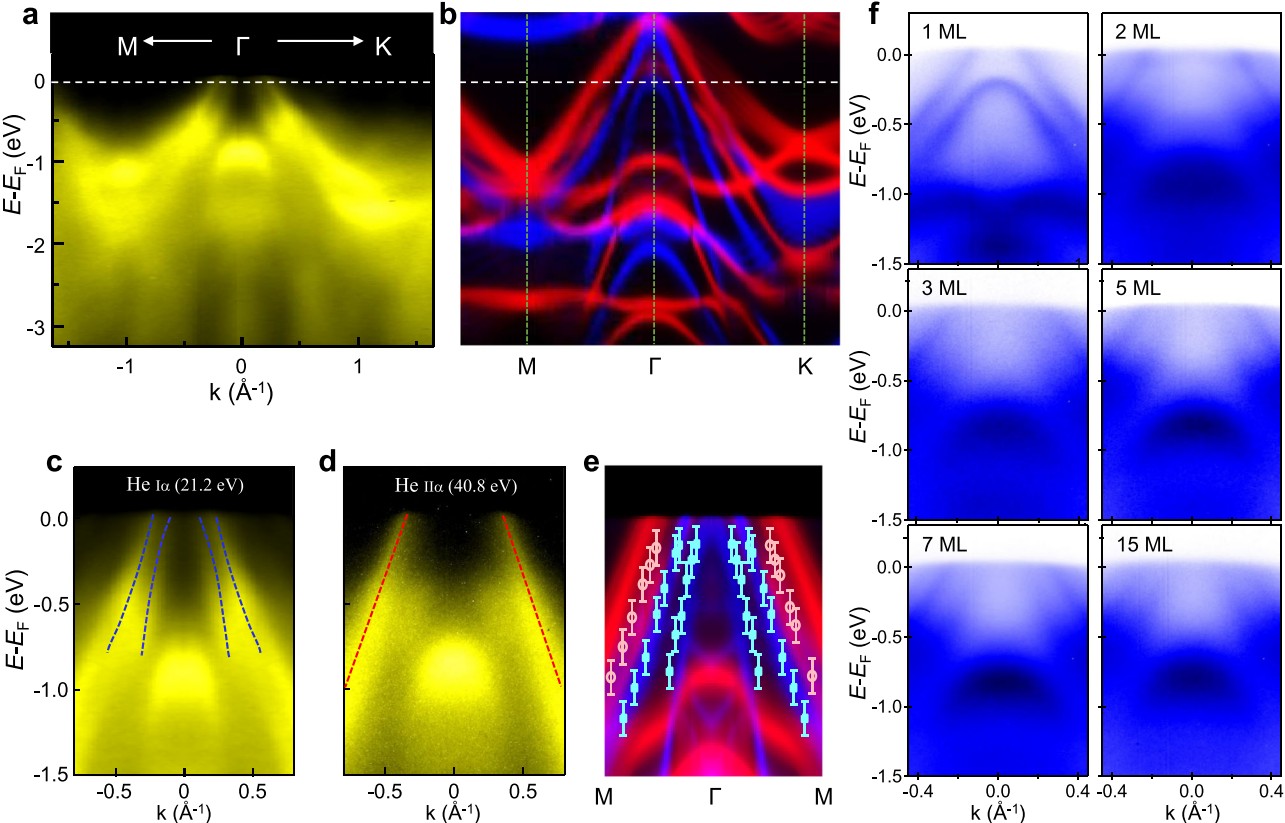

**Fig. 5 Band structure of CrTe₂ ultrathin films. a, b** Plots of valence-band dispersion (**a**) and the first-principles calculations (**b**) of 7 ML CrTe₂ with the inclusion of spin polarization along the high symmetry direction M-Γ-K. The minority and majority spin bands are plotted in red and blue colors, respectively. The major features seen in the left panel are well reproduced in the right one. **c–e** Comparison of the valence-band dispersion near the Fermi level taken by He Iα (21.2 eV) (**c**), He IIα photons (40.8 eV) (**d**) with theoretical bands (**e**) along the high symmetry direction M-Γ-M. The blue and red dashed lines indicate the position of hole pockets measured by He Iα and He IIα photons, respectively. The light blue/red markers represent the positions of MDC peaks. The error bars represent uncertainties in locating peak positions. **f** ARPES intensity maps of 1 ML, 2 ML, 3 ML, 5 ML, 7 ML, and 15 ML, respectively. The spectra of various thicknesses were taken along the high symmetry direction M-Γ-M.

the film thickness, the Fermi level moves towards the valence band with the band shape invariant. To understand the thickness-dependent electronic structure, we carried out first-principles calculations of 1T-CrTe₂ with different thicknesses (see Supplementary Fig. 18). There is an excellent agreement between our experiment and theory. In particular, the hole-like band near $E_F$ and a relatively flat Cr 3*d* orbital band are similar to that of calculated 1T-CrTe₂ with the inclusion of spin polarization. For the 1 ML film, the two parabolic hole pockets are well reproduced by the majority spin projections of the bands, which highlights the FM nature. These results demonstrate that the epitaxial 1 T structure and ferromagnetism have been established since 1 ML deposition, in line with the corresponding STM images. The layer-by-layer growth mode of the CrTe₂ ultrathin films enables us to further explore the interplay between electronic structure and extraordinary magnetic properties on the basis of thin-film electronic devices.

To summarize, we have successfully synthesized high-quality mono- to few-layer CrTe₂ via MBE method, for the first time. The epitaxial CrTe₂ ultrathin films with thickness up to 7 ML possess room-temperature intrinsic ferromagnetism, large magnetic moments (~3 $\mu_B$/atom), strong perpendicular anisotropy and magnetic spin-split band structure. The high $T_C$ is preserved with the thickness down to one ML due to the strong magnetic anisotropy and the weak interlayer coupling. The FM CrTe₂ films can be employed as a spin injector when hybridized with other 2D materials such as topological insulator and topological

semimetals for exploring novel spin physics. At the same time, this work provides a tremendous potential for the future 2D magnet-based spintronics technologies, as the films can readily reach wafer size with MBE growth technique.

## Methods

**Growth of CrTe₂/bilayer graphene/SiC(0001) heterostructures.** CrTe₂ thin films were grown on a bilayer graphene/SiC substrate in an integrated MBE-STM ultrahigh vacuum (UHV) system with base pressure below $2 \times 10^{-10}$ mbar. The bilayer graphene was prepared by annealing a 6H SiC(0001) substrate at 1150 °C for 20 s and repeating 30 times. Then, high-purity Cr and Te were evaporated from an electron-beam evaporator and a standard Knudsen cell, with flux of 0.1 Å/min and 6 Å/min, respectively. The temperature of substrate was kept at 375 °C during the growth. The deposition rate of CrTe₂ was ~0.73 Å/min as monitored by a quartz oscillator. In order to protect the thin film from contamination and oxidation during XRD, SQUID, XAS, and XMCD measurements, a Te capping layer (~5 nm) was deposited on sample surface after growth.

**Characterizations.** High-resolution XRD was performed using MoK$_{\alpha1}$ radiation (0.70926 Å) which was obtained from a flat perfect crystal Ge monochromator that produced a line beam having angular divergence of in the scattering plane and out of the scattering plane. The measurements were performed by specular reflection and the data were modeled using the reflection amplitudes from the substrate, graphene layers, layers of CrTe₂, and its structure factor. The TEM samples were prepared by a lift-out method in a ThermoFisher Scientific Scios focused ion beam (FIB) instrument at room temperature, and imaged in the ThermoFisher Scientific G2 Tecnai F30 FEG high resolution TEM operated at 300 kV. The SiC substrate was tilted to the [100] zone axis and the lattice fringes from both the graphene and the SiC can be clearly resolved. Great care has been taken to reduce the beam damage on the thin film samples both during the FIB lift out and during the sample tilting and high-resolution image acquisition process. The magnetization

measurements were performed by using a Quantum Design SQUID magnetometer with magnetic field up to 7 T.

**X-ray absorption spectroscopy and magnetic circular dichroism**. The measurements were performed on beamline I10 at Diamond Light Source, UK, with 100% circularly polarized X-ray perpendicular to the sample plane and parallel to the magnetic field. XAS measurements with total electron yield (TEY) mode were carried out from 5 K to 300 K. By flipping the X-ray helicity at fixed magnetic field of 1 T, we obtained XMCD by taking the difference of XAS, $\sigma^+$- $\sigma^-$.

**Angle-resolved photoemission spectroscopy and scanning tunneling microscopy**. After the growth, the $CrTe_2$ films were in-situ transferred under ultra-high vacuum to the ARPES stage. ARPES measurements were performed at 107 K using a SPECS PHOIBOS 150 hemisphere analyzer with a SPECS UVS 300 helium discharge lamp (He I$\alpha$ = 21.2 eV and He II$\alpha$ = 40.8 eV). The energy resolution is 40 meV under 107 K. The size of the beam spot on the sample was ~1.5 mm. We didn't find any change in the observed ARPES spectra when changing the beam position on the sample surface (~ 5 mm × 4 mm), indicative of the homogeneity of grown samples. The topography of the sample surface was mapped in-situ by an Aarhus STM housed in the growth chamber.

**First-principles calculations**. First-principles calculations with DFT were performed by using the Vienna ab Initio Simulation Package (VASP) package. We used the Perdew–Burke–Ernzerhof (PBE) form of the exchange correlation functional. All the calculations were performed with a plane-wave cut-off energy of 300 eV on the $11 \times 11 \times 1$ Monkhorst-Pack k-point mesh. The super cell includes $CrTe_2$ layers with varying thicknesses and a vacuum layer of about 20 Å, in order to avoid interactions between the neighboring slabs. $CrTe_2$ with an in-plane lattice constant of 3.81 Å was used. The atomic positions and the out-of-plane lattice constant were optimized by the conjugate gradient method. Calculations of the band structures were performed with the inclusion of SOC.

## Data availability
The authors declare that the main data supporting the findings of this study are available within the article and its Supplementary Information files. Extra data are available from the corresponding authors upon reasonable request.

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

## Acknowledgements

This work is supported by the National Key Research and Development Program of China (No. 2016YFA0300803, No. 2017YFA0206304), the National Basic Research Program of China (No. 2014CB921101), the National Natural Science Foundation of China (No. 61427812, 11774160, 11574137, 61474061, 61674079, 11904174), Jiangsu Shuangchuang Program, the Natural Science Foundation of Jiangsu Province of China (No. BK20140054, BK20190729), NUPTSF (Grant No. NY219024), the Natural Science Foundation of the Jiangsu Higher Education Institutions of China (19KJB510047), UK EPSRC (EP/S010246/1), leverhulme Trust (LTSRF1819\15\12), and Royal Society (IEC \NSFC\181680). G.B. is supported by the US National Science Foundation (NSF-DMR#1809160). Work of David Singh is supported by the U.S. Department of Energy, Basic Energy Sciences, Award Number DE-SC0019114. Diamond Light Source is acknowledged to I10 under proposal MM22532. T.-R.C. is supported by the Young Scholar Fellowship Program from the Ministry of Science and Technology (MOST) in Taiwan, under a MOST grant for the Columbus Program MOST108-2636- M-006-002, National Cheng Kung University, Taiwan, and National Center for Theoretical Sciences, Taiwan. This work is also partially supported by the MOST, Taiwan, Grant MOST107-2627-E-006-001 and by Higher Education Sprout Project, Ministry of Education to the Headquarters of University Advancement at National Cheng Kung University (NCKU).

## Author contributions

Y.X., G.B., R.Z., and L.H. planned the project. X.Z. and Q.L. synthesized CrTe$_2$ thin films. Q.L., X.Z., and J.C. conducted the ARPES and STM experiments and analyzed the data. W.L., J.S., W.N., and J.D. performed XMCD and SQUID measurements and analyzed the data. Q.L., S.-W.L., T.-R.C., and D.J.S. did the DFT calculations. P.M. and M.V. conducted XRD measurements. X.H. performed the TEM characterizations. X.Z., Q.L., D.J.S., and G.B. wrote the paper. All the authors discussed the results and commented on the manuscript.

## Competing interests

The authors declare no competing interests.
