## [Peer Review File · Nature Communications]

Reviewers' Comments:

Reviewer #1:

Remarks to the Author:

Xiaoqian Zhang et al. report on a multi-characterisation of ultra thin van der Waals CrTe₂ films. The films have been grown by MBE on graphene/SiC. This system is a very promising one in terms of (close to) room temperature ferromagnetic, with possible implications for fundamental understanding of low-dimensional magnetism and for spintronics applications.

The authors have used a large set of methods to study their system, including MBE, STM, XRD, SQUID, XMCD-XAS, ARPES measurements, and DFT calculations. This is certainly one of the strong points of the manuscript. The authors provide strong pieces of evidence that the material is indeed magnetic close to room temperature, with a large magnetic anisotropy and significant coercivity, even in the very few-layer thickness regime. Compared to recent works cited in the manuscript, which addressed thicker (still thin) films, this is certainly an important step.

I believe that the work presented here is of potential strong interest for the large communities working on 2D materials or spintronics. However, I think that the manuscript needs to be improved in several respects. I provide a list of comments/suggestions/questions below.

1. The introduction is generally hard to follow, as it invokes rather advanced condensed matter physics concepts, and overall lacks clarity. Some recent reviews, for instance the one published by Gibertini and coworkers, have managed to introduce key concepts in a rather clear way. The authors might find inspiration in such reviews.

2. Still in the introduction, the authors make a strong point about the role of dimensionality on the "robustness of 2D ferromagnetic order". Given the thickness of the materials this is certainly an important point, but this cannot be the only one: almost all parent 3D compounds that have been exfoliated as 2D materials already have a low Curie temperature (in the bulk form), for instance, and this is not, at least at first sight, related to a dimensionality issue. I think it is important to convey this idea to the reader, and I advise the authors to include a discussion in the introduction.

3. The writing style should be improved in many instances: I find that some sentences are very lengthy, many others are not grammatically correct, the style is sometimes overly emphatic (e.g. outstanding, incredibly significant, particularly significant), and several formulations are obscure. I recommend the authors to thoroughly revise the text of their manuscript in this respect.

4. It is unclear to me whether the STM or XRD characterisations presented by the authors can make any difference between 1T and 2H CrTe₂. This should be clarified in the text (although I understand that other pieces of evidence indeed suggest the formation of 1T CrTe₂).

5. I am not sure that the sentence "showing that the FM order of the CrTe₂ is insensitive to the thickness of samples" is actually consistent with the data, and I am not even sure to understand the meaning of this sentence.

6. Regarding magnetic characterisations and DFT calculations, I would suggest that the authors more thoroughly compare their results to those reported by Purbawati et al. (ref. 20) and Freitas et al. (ref. 21) respectively.

7. The authors should show SQUID data acquired with the bare SiC/graphene substrate. This is an important piece of information needed to rule out any possible magnetic contribution from possible magnetic impurities in the substrate. I agree that XMCD should not be sensitive to such effects, but given the numerous debates on whether some 2D materials are magnetic or not, I suggest extra care in the presentation and interpretation of the SQUID data.

8. It is unclear whether the samples investigated with XRD were capped with Te. It is also unclear whether the Te capping layer was later (e.g. for ARPES, XAS and XMCD measurements) desorbed under vacuum.

9. In the sentence "which is also confirmed by the calculated density of states (Fig. S6)", I guess the authors mean "Fig. S7". I have the feeling (but might be wrong...) that in several instances, the supplementary figures are not properly called in the main text. Please check.

10. Regarding the ARPES measurements and the possible match with DFT simulations performed for thin 1T CrTe₂ flakes, I have several questions. The authors write "The observed metallic band further confirms the trigonal phase of the CrTe₂ thin films". However, Fig. S9 for instance shows that both the 1T and the 2H phases are predicted metallic (at least in the bulk form). Actually, I find it very difficult to conclude that the observed electronic band structure resembles the one predicted based on DFT calculations. The bands are actually not nicely resolved, and there are many features that do not really look alike what is calculated (Fig. 4a/b). What about a calculation for the 2H phase of a thin CrTe₂ film (not for bulk, like in Fig. S9)? The number of layers seems an important parameter, and systematic calculations for different layer number might be relevant here. By the way, I might have missed the information, but I did not figure out what the layer number is for Fig. 4a. All this needs to be clarified and maybe discussed with more care.

111. Too few details are given about how the DFT calculations and electronic band structure was simulated.

Reviewer #2:

Remarks to the Author:

Two-dimensional van der Waals magnet is a new class of materials, being recognized as a unique platform for studying the two dimensional magnetism and also an important building block of van der Waals interfaces for novel spintronic functionalities.

In this work, authors report the epitaxial growth of a new 2D van der Waals magnet CrTe₂ and its magnetic/electronic properties. The results (High transition temperature, perpendicular magnetic anisotropy, and layer dependent magnetic/electronic properties) are clearly indicates the potential of this material.

I feel that results are worth publishing in Nature Communications. However, there are several unclear points and I think the authors should further clarify them.

1. Although authors mentioned about the stoichiometry of grown films and excluded the possibilities of CrTe, Cr₂Te₃, and Cr₅Te₈, it requires more careful discussion to conclude the stoichiometry. It is known that MBE-grown sample can have its unique stoichiometry (partially intercalated one, for example), which is different from bulk and can even show the thickness dependence. In order to unambiguously clarify it, authors should show the cross-sectional TEM image of their samples with several different thicknesses.

2. I am confused about the valence number of Cr. Considering the stoichiometry of CrTe₂, it seems that Cr⁴⁺ (and Te²⁻) is reasonable. Why authors can conclude Cr³⁺?

3. Related with the above comment, can Cr-Te bond be regarded as covalent bond or ionic bond?

4. I think it is impossible to determine the space group from out-of-plane XRD (line 133-136 in page 6).

5. For monolayer samples, authors only show the ARPES data. They should show the magnetization and XMCD data for monolayer sample. Because it seems that transition temperature of 3 ML samples is around 250 K (Fig. 2 a) and that of monolayer might be lower than it, the word "monolayer" in the title is misleading.

6. They only show the ARPES data at 107 K. More detailed discussion about the temperature dependence of ARPES data (temperature dependent spin splitting value) is required.

7. What is the mechanism of perpendicular magnetic anisotropy? Is it consistent with the bulk

CrTe₂?

8. In Fig. 2a, it is written that magnetic field is applied in-plane. It is correct? If so, it is better to replace it with that under the out-of-plane magnetic field.

9. 15ML data in Fig. 2 a shows slight decrease at low temperature? What is the origin of this strange behavior?

10. Authors claim that orbital magnetization is almost negligible. If so, why magnetic anisotropy appears?

Reviewer #3:

Remarks to the Author:

Dear Authors,

In this manuscript, the authors report the observation of room temperature ferromagnetism in CrTe₂ thin films. The manuscript is organized with interesting experimental results about structural characterization (Fig. 1 and S1), magnetization (Fig. 2), anomalous Hall effect (Fig. S2), circular dichroism (Fig. 3 and S3), and ARPES with calculation (Fig. 4 and S4-S9). I totally agree that the successful fabrication of high quality CrTe₂ ultrathin films is quite important advancement. By applying the interface formation, the spintronic function can be expected to work at room temperature. However, in previous studies, the room temperature ferromagnetism has been observed in bulk and exfoliated thin flakes of CrTe₂. Of course, XMCD and ARPES are also important experimental results to understand the electronic structures of the CrTe₂ films. Considering these points, for my sense, it is difficult to find new significant achievements in the present manuscript.

I provide some comments to improve the manuscript.

1, appropriate definition should be used.

The definition in this manuscript should be carefully improved, for example "intrinsic ferromagnetism" in the title and "intrinsic 2D ferromagnetism" in the abstract, are described but no discussion is provided in main text. What does "intrinsic" mean? How did the authors conclude "2D" ferromagnetism? The meaning of "2D magnet" is a 2D material possessing magnetism. It is generally accepted. However, the meaning of "Intrinsic 2D ferromagnetism" is completely different from that of 2D magnet. The dimensionality of the ferromagnetism has to be examined.

If the films have less defect concentration like bulk single crystals, the observed ferromagnetism is supported by similar origin in the bulk crystals. It is great that the authors explain the basis for what is intrinsic and extrinsic magnetism.

Room temperature ferromagnetism is obtained in only 7 and 15 ML films. These thicknesses are comparable to that in the previous studies of exfoliated flakes. Please describe the results with fair comparison. The plot of Curie temperature as a function of thickness is quite meaningful, because the analysis for Curie temperature of each film is not clearly explained.

2, magnetic anisotropy should be discussed.

In this manuscript, the perpendicular magnetic anisotropy (PMA) is claimed. This feature is different from that in bulk and exfoliated flakes. However, the reason is not discussed well. I recommend to discuss the reason why PMA is obtained in the films.

Although Fig. 2a shows temperature dependence of in-plane magnetization, temperature dependence of perpendicular magnetization should be also plotted if the authors try to claim the perpendicular magnetic anisotropy. In Fig. 2a, the small magnetization actually remains at 300 K with weak temperature dependence. Judging from this two-step-like temperature dependence, it is natural that two magnetic origins play a role in this film, in other words, weak dependent component around 300K and strong dependent component at low temperature region. The authors should clarify the two origins in the M-T curve by analysis for usual magnet with critical exponent beta term.

The authors claim that the saturation magnetization value of the films is consistent to the value in the bulk. However, it is apparent that the magnetization value does not linearly follow the

thickness in Fig. 2a. The magnetization value should be calculated and presented by number of moment for Cr atoms ($\mu\text{B}/\text{Cr}$) as discussed in Fig. 3d. Then, the values can be compared to that in bulks and that evaluated by XMCD. If the magnetization values in the films depends on the thickness, the origin should be discussed.

3, relationship between magnetization and anomalous Hall effect.

Probably the authors already notice, anomalous Hall (AH) effect reflects the perpendicular magnetization. In Fig. S2, the AH resistance for 3 ML film apparently disappears at 250 K, indicating no or rather weak perpendicular magnetization at 250 K. The AHE is measured with 3-layers film so that the M-T curve for 3ML shown in Fig. 2a seems consistent. However, M-H curve in Fig. 2b is measured with 7ML film. To remove such ambiguity, the authors should show all data set to compare Curie temperature and coercive field between AHE and magnetization for 3ML, 5 ML, and 7 ML, if the authors try to claim that the ferromagnetic order can be obtained in ultrathin 3 ML with perpendicular magnetic anisotropy. Perpendicular magnetic anisotropy should be concluded by the discussion and experimental results on the comparison of M-H curves in-plane and out-of-plane. In the present manuscript, the data set of M-H curves for 3 ML and 5 ML is not provided. The authors have to show these experimental evidences.

In addition, what happen in 2 ML and 1 ML film ? The authors describe "indicating a negligible dimensionality effect" in page 5, but the Curie temperature and magnetization decreases with decreasing thickness. Please reconsider this point. I agree that it is difficult to detect magnetization in such ultrathin films by SQUID but they may detect it by XMCD. If the authors have measured those films, it is great to provide the discussion for 2 ML and 1 ML. By ARPES measurement, electronic bands are well discussed with increasing thickness. Because ferromagnetism is observed above 3 ML, did the authors detect the critical difference between 1,2 ML and 3 ML in terms of electronic structures? The relationship between electronic structure and magnetism against thickness variation is an interesting topic in such 2D layered magnetic compounds.

4, Temperature dependence of magnetization in Fig. 3d is apparently different trend with that in Fig. 2a. Fig. 3d shows monotonic increase with decreasing temperature. However, 7 ML data in Fig. 2a steep increase around 250 K, then keep constant at low temperature. This difference should be discussed. If this difference comes from the measurement configuration, in-plane or out-of-plane, the authors should provide temperature dependence of out-of-plane magnetization. If this comes from thickness difference or measurement method, it should be explained.

5, PMA is compared with K_u and saturation magnetization. K_u should be explained by equation. The magnetization in CrTe_2 is rather small compared to other metal ferromagnets with PMA. Degree of PMA should be discussed on the same measure. For my sense, "strong PMA" is not appropriate for the observed data of CrTe_2 in this manuscript because the value is not so large. If the authors try to claim the "strong PMA", please provide the values for famous strong PMA metal ferromagnets to compare.

6, please check carefully following minor points.

6-1, "doping a FM host with specific elements" in page 3, a word of FM means ferromagnetic. What did the word "a FM host" mean?

6-2, "modulating the magnetic order via electrolyte gating" in page 3, gating can modulate the charge density and/or electric field at the interface, resulting in the variation of magnetism. The word of "magnetic order" has some meanings, which may confuse. Please revise carefully.

6-3, Refs.10-12 are really appropriate references for the enhancement of FM order by proximity effect ?

6-4, "FM order of the CrTe_2 is insensitive to the thickness" in page 6 is difficult to agree, because the judgement whether sensitive or insensitive depends on the system. The plot for T_c of film/ T_c of bulk as a function of thickness for CrTe_2 and $\text{Cr}_2\text{Ge}_2\text{Te}_6$ seem meaningful to compare.

6-5, What does "quantum thickness regime" in page 7 mean? How did the authors define and evaluate the quantum feature?

6-6, "large magnetic moments" in page 12 is not discussed with the exact values for the films in main text. It is difficult to judge the large or small moment.

6-7, "the thickness down to 3 ML due to the strong magnetic anisotropy" in page 12 is not supported by experimental results, because the M-H curves of in-plane and out-of-plane for 3 and 5 ML films are not provided in the present manuscript. PMS should be judged by comparison of M-H curves of in-plane and out-of-plane.

6-8, Growth rate and provided flux rate should be addressed for general readers.

6-9, There is a paper not referred in this manuscript,

Room temperature ferromagnetism in ultra-thin van der Waals crystals of 1T-CrTe₂, Nano Research. Doi.org/10.1007/s12274-020-3021-4.

Response to referee reports

First of all, we thank all three reviewers for their efforts, and the review comments/suggestions which we find very helpful and constructive. We have taken them fully into account in our revised paper. We truly hope that you will find it now worthy of publication in *Nature Communications*.

Below we provide our point-to-point responses to all of the review comments.

Reviewer #1 (Remarks to the Author):

Comments:

Xiaoqian Zhang et al. report on a multi-characterization of ultra thin van der Waals CrTe₂ films. The films have been grown by MBE on graphene/SiC. This system is a very promising one in terms of (close to) room temperature ferromagnetic, with possible implications for fundamental understanding of low-dimensional magnetism and for spintronics applications.

The authors have used a large set of methods to study their system, including MBE, STM, XRD, SQUID, XMCD-XAS, ARPES measurements, and DFT calculations. This is certainly one of the strong points of the manuscript. The authors provide strong pieces of evidence that the material is indeed magnetic close to room temperature, with a large magnetic anisotropy and significant coercivity, even in the very few-layer thickness regime. Compared to recent works cited in the manuscript, which addressed thicker (still thin) films, this is certainly an important step.

I believe that the work presented here is of potential strong interest for the large communities working on 2D materials or spintronics.

Reply: We thank the reviewer for his/her positive comments on the manuscript and the fact that he/she thinks that our work is of potential strong interest for the large communities working on 2D materials or spintronics.

Comments: However, I think that the manuscript needs to be improved in several respects. I provide a list of comments/suggestions/questions below.

1. The introduction is generally hard to follow, as it invokes rather advanced condensed matter physics concepts, and overall lacks clarity. Some recent reviews, for instance the one published by Gibertini and coworkers, have managed to introduce key concepts in a rather clear way. The authors might find inspiration in such reviews.

Reply: We thank the reviewer for this kind suggestion.

Revision: Based on the reviewer's suggestion, we have rephrased the introduction in the revised manuscript on page 3.

"In a three-dimensional (3D) system, the magnon density of states are consecutive and chiefly determined by exchange interactions. Therefore, a magnetic phase transition could occur at a finite temperature. On the contrary, the long-range magnetic order in 2D systems can be destroyed by thermal fluctuations, according to the Mermin-Wagner theorem^{1,2}. Considering the magneto-anisotropy in 2D ferromagnets, it opens up a large spin-wave excitation gap and quenches thermal fluctuations³⁻⁹, thereby stabilizing the long-range magnetic order in 2D regime. Opposed to defect or dopant induced magnetism, the ferromagnetism occurring in a stoichiometric compound is defined as intrinsic ferromagnetism."

2. Still in the introduction, the authors make a strong point about the role of dimensionality on the "robustness of 2D ferromagnetic order". Given the thickness of the materials this is certainly an important point, but this cannot be the only one: almost all parent 3D compounds that have been exfoliated as 2D materials already have a low Curie temperature (in the bulk form), for instance, and this is not, at least at first sight, related to a dimensionality issue. I think it is important to convey this idea to the reader, and I advise the authors to include a discussion in the introduction.

Reply: We thank the reviewer for this valuable comments. This is indeed an important point. We agree with the reviewer that most of parent 3D compounds which could be exfoliated as 2D materials already have a relatively low Curie temperature (T_C) in the bulk form, such as CrI_3 (61 K)¹, $\text{Cr}_2\text{Ge}_2\text{Te}_6$ (68 K)², CrCl_3 (17 K)³ and CrSiTe_3 (33 K)⁴. For 3D compounds, the T_C is predominately determined by exchange interactions², and is insensitive to small single-ion anisotropies or small fields. Moreover, the interlayer exchange interactions in van der Waals (vdW) crystals is 2–3 orders of magnitude weaker than that of traditional metals². Thereby, the T_C of 2D ferromagnets in the bulk form is usually lower than that of traditional 3D materials⁵.

Revision: According to the reviewer's suggestions, we have included the discussion of low T_C of 2D ferromagnets in the bulk form in the introduction part of the revised manuscript on page 3.

"It mainly results from the enhanced spin fluctuation in reduced dimensions or the relatively weak exchange interactions. Note that the interlayer bonding strength in vdW compounds is 2–3 orders of magnitude weaker than that of traditional 3D materials, which leads to a low T_C in the bulk form already."

3. The writing style should be improved in many instances: I find that some sentences are very lengthy, many others are not grammatically correct, the style is sometimes overly emphatic (e.g. outstanding, incredibly significant, particularly significant), and several formulations are obscure. I recommend the authors to thoroughly revise the text of their manuscript in this respect.

Reply: We appreciate the careful reading and suggestions to improve the English and presentation in the manuscript. We have gone through carefully the entire manuscript and improved the language throughout.

4. It is unclear to me whether the STM or XRD characterizations presented by the authors can make any difference between 1T and 2H CrTe₂. This should be clarified in the text (although I understand that other pieces of evidence indeed suggest the formation of 1T CrTe₂).

Reply: We thank the reviewer for this nice advice. With STM or XRD characterizations, we can distinguish the structures of 2H and 1T CrTe₂. As shown in Fig. R1, the 2H structure is composed of a hexagonal lattice with 2 formula units (f.u.) per unit cell, whose atom planes are in the AbA BaB stacking sequence, belonging to $P6_3/mmc$ space group ($a = 3.49 \text{ \AA}$, $c = 13.64 \text{ \AA}$)⁶. In contrast, the 1T structure is composed of a hexagonal lattice with 1 f.u. per unit cell with ABC stacking sequence, belonging to $P\bar{3}m1$ space group ($a = 3.79 \text{ \AA}$, $c = 5.94 \text{ \AA}$)⁶. Therefore, these lattice constants can be distinguished by STM and XRD. We found that the step height and in-plane lattice constant taken from STM measurements are 6.14 \AA and 3.81 \AA , respectively, manifesting the 1T phase of as-grown CrTe₂ films. Moreover, according to XRD characterization, the lattice constant $c = 6.13 \text{ \AA}$, identical with STM results. Both XRD and STM measurements indicate that the lattice parameters are consistent with 1T phase, not 2H.

Fig. R1 Lateral view of CrTe₂ with 2H and 1T structures. The unit cells are represented in dashed lines.

Revision: We have added a discussion of the STM and XRD results about 1T and 2H phase in the revised manuscript on page 6.

“A typical X-ray diffraction (XRD) 2θ - ω scan was employed to further identify the crystal structure (Fig. 1d). The diffraction pattern with perpendicular constant $c = 6.13 \text{ \AA}$ is matched to the (001) crystal planes of the standard 1T-type hexagonal structure ($a = 3.79 \text{ \AA}$, $c = 5.94 \text{ \AA}$), rather than 2H phase ($a = 3.49 \text{ \AA}$, $c = 13.64 \text{ \AA}$)⁶. With STM,

TEM and XRD characterizations, the formation of CrTe₂ films with 1T phase and their single-crystalline nature has been confirmed.”

5. I am not sure that the sentence "showing that the FM order of the CrTe₂ is insensitive to the thickness of samples" is actually consistent with the data, and I am not even sure to understand the meaning of this sentence.

Reply: We agree with the reviewer that this statement might be misleading, and should be corrected. In order to investigate the dimensionality effect of the ferromagnetism in CrTe₂, we provide additional XMCD measurement on 1 ML CrTe₂ film, which displays a T_C of ~200 K. Please refer to Fig. R9 for detailed temperature dependent XMCD results. The thickness dependent normalized T_C (T_C/T_{C_bulk}) is summarized in Fig. R2. Compared with other commonly investigated 2D magnets^{2,7}, the T_C of CrTe₂ decreases slightly with reducing film thickness, demonstrating the robustness of ferromagnetism in CrTe₂ thin films.

Fig. R2 T_C (normalized to bulk T_C for the particular material) as a function of the number of layers. Data for Fe₃GeTe₂ adapted from ref. ⁷; for Cr₂Ge₂Te₆, ref. ².

Revision: In the revised manuscript on page 11, we have changed this sentence as “*In order to investigate the dimensionality effect of the ferromagnetism in CrTe₂ stemming from thermal fluctuation, we plot the thickness dependent T_C (Fig. 4d). The T_C of CrTe₂ decreases slightly with reducing the film thickness compared with other commonly investigated 2D magnets, such as Cr₂Ge₂Te₆² and Fe₃GeTe₂⁷ (Supplementary Fig. 8), demonstrating the robustness of ferromagnetism in the epitaxially grown CrTe₂ thin films.*”

6. Regarding magnetic characterizations and DFT calculations, I would suggest that the authors more thoroughly compare their results to those reported by Purbawati et al. (ref. 20) and Freitas et al. (ref. 21) respectively.

Reply: Thanks for the nice advice. In terms of magnetic properties, CrTe₂ few-layer films show a strong perpendicular magnetic anisotropy (PMA). It is different from the

bulk CrTe₂ reported by Freitas *et al.*⁸ and exfoliated CrTe₂ flakes (thicker than 10 nm) reported by Purbawati *et al.*⁹, which possess in-plane easy axis. Despite these distinct results, we attribute them to the thickness-dependent magnetic anisotropy. The origin of magnetic anisotropy generally originates from three mechanisms, shape anisotropy^{10,11}, magneto-crystalline anisotropy¹² and anisotropy of exchange interactions¹¹. The shape anisotropy is due to the magnetostatic or dipole interactions, which lead to a preferential in-plane anisotropy for thin films¹¹. The PMA in CrTe₂ thin films is contrary to it. Thereby, the possibility of shape anisotropy can be ruled out.

The anisotropy of exchange interactions can be verified by performing DFT calculations of exchange interactions using the frozen magnon approach¹². Fujisawa *et al.* found that the ferromagnetic (FM) Cr-Cr intra-sublattice exchange interaction dominates in CrTe₂ thin films, and the total energy minima for CrTe₂ occurs at perpendicular direction¹². In this vein, the magnetic moments of CrTe₂ thin films are driven to perpendicular direction due to the anisotropy of exchange interactions.

The thickness dependent magnetic anisotropy suggests that the lowered symmetry at the interface plays an important role in determining the PMA in CrTe₂ thin films. As the magnetic film thickness approaches a few nm, interfacial magnetism and inversion symmetry breaking give rise to intriguing phenomena such as PMA¹³. This is a consequence of spin-orbit interactions, *i.e.*, magneto-crystalline anisotropy, that apparently have a stronger effect in the more anisotropic film limit^{10,14,15}. Both experimental results and DFT calculations indicate that, the magnetic moments of CrTe₂ thin films are driven to the perpendicular direction, due to the magneto-crystalline anisotropy and the anisotropy of exchange interactions.

Regarding DFT calculations, according to the orbital and surface projection analysis of the band structure of 7 ML CrTe₂ in this work, the metallicity is a consequence of the hybridization of Te-5*p* and Cr-3*d* orbitals crossing the Fermi level at the center of the Brillouin zone (Fig. S11), which is confirmed by the calculated density of states (Fig. S12). The hybridization of Te and Cr bands is also verified with DFT calculations (including band structure and density of states) made by Freitas *et al.*⁸. Moreover, the magnetic splitting of ~2-3 eV in bulk CrTe₂ is also predicted, suggesting a FM ground state. Similarly, the splitting of majority and minority bands in CrTe₂ films corroborates the FM ground state, which highlights the unique interplay of ferromagnetism and electronic structure in CrTe₂.

Revision: We have included the discussion of magnetic characterizations on page 8 and DFT calculations with those reported by Purbawati *et al.* and Freitas *et al* on page 12.

“The strong PMA in CrTe₂ few-layer films is different from bulk CrTe₂⁸ and exfoliated flakes (thicker than 10 nm)¹⁶ which have an in-plane easy axis. Here, the thickness dependent magnetic anisotropy suggests that the reduced symmetry at the interface plays an important role in determining the PMA in CrTe₂ thin films¹⁵. As the magnetic

film thickness approaches a few nm, the interfacial magnetism and inversion symmetry breaking give rise to the PMA¹³. This is a consequence of spin-orbit interactions, i.e., magneto-crystalline anisotropy, that apparently have a stronger effect in the more anisotropic film limit^{10,14,15}. In addition, based on density functional theory (DFT) calculations, it has been found that the FM Cr-Cr intrasublattice exchange interactions dominate in CrTe₂ thin films, and the total energy minima is at perpendicular direction¹². In general, the magnetic moments of CrTe₂ thin films are aligned in the perpendicular direction, due to the magneto-crystalline anisotropy and the anisotropy of exchange interactions.”

“The hybridization of Te and Cr bands is also verified with DFT calculations (including band structure and density of states) made by Freitas et al⁸.” “The magnetic splitting in bulk 1T-CrTe₂ is also predicted by Freitas et al⁸, suggesting a FM ground state.”

7. The authors should show SQUID data acquired with the bare SiC/graphene substrate. This is an important piece of information needed to rule out any possible magnetic contribution from possible magnetic impurities in the substrate. I agree that XMCD should not be sensitive to such effects, but given the numerous debates on whether some 2D materials are magnetic or not, I suggest extra care in the presentation and interpretation of the SQUID data.

Reply: We thank the reviewer for the nice suggestion. Based on this comment, we did the extra control experiments of bare SiC/graphene substrate. As exhibited in Fig. R3, the field dependent magnetization of SiC/graphene substrate at 20 K shows a typical diamagnetic behavior clearly. Instead, the magnetic response of 7 ML CrTe₂ film on SiC/graphene substrate displays an obvious FM hysteresis loop superposed on a diamagnetic background. Therefore, the possibility of magnetic contribution from magnetic impurities in the substrate can be ruled out.

Fig. R3 Eliminating the possible magnetic contribution from the substrate. (a) The out-of-plane field dependent magnetization of SiC/graphene substrate taken at 20 K shows a linear relationship, demonstrating a typical diamagnetic behavior. (b) Well-defined magnetic hysteresis loop of 7 ML CrTe₂ film without subtracting the diamagnetic

background from substrate, indicating the intrinsic ferromagnetism of CrTe₂ thin films.

Revision: We have added the data of the control experiment in the supplementary materials (Fig. S4), as well as the discussion of magnetic contribution from substrate in the manuscript on page 7.

“Control experiments on the field dependent magnetization of SiC/graphene substrate show a typical diamagnetic behavior (Supplementary Fig. 4). Therefore, the possibility of magnetic contribution from magnetic impurities in the substrate can be ruled out.”

8. It is unclear whether the samples investigated with XRD were capped with Te. It is also unclear whether the Te capping layer was later (e.g. for ARPES, XAS and XMCD measurements) desorbed under vacuum.

Reply: Thank you for the comment. The samples investigated with XRD were capped with 5 nm Te to prevent the contamination and oxidation. Few-layer 2D FM materials are going to degrade under ambient atmosphere⁷, and the capping layers are indispensable. For the ARPES measurements, CrTe₂ samples were in-situ transferred under ultra-high vacuum to the ARPES chamber after finishing the film growth. Therefore, there is no need to evaporate a capping layer. For the XAS and XMCD measurements performed on beamline, CrTe₂ samples were capped with 5 nm Te to prevent the oxidation and environmental doping during transport to the synchrotron facility. Since the escaping length of photoelectrons by XAS and XMCD characterization in TEY mode is ~10 nm¹⁷, Te capping layer doesn't need to be desorbed under vacuum.

Revision: We have added the experimental details about capping layers in the revised manuscript on page 15 and 16.

“In order to protect the thin film from contamination and oxidation during XRD, SQUID, XAS and XMCD measurements, a Te capping layer (~5 nm) was deposited on sample surface after growth.” “After the growth, the CrTe₂ films were in-situ transferred under ultra-high vacuum to the ARPES stage.”

9. In the sentence "which is also confirmed by the calculated density of states (Fig. S6)", I guess the authors mean "Fig. S7". I have the feeling (but might be wrong...) that in several instances, the supplementary figures are not properly called in the main text. Please check.

Reply: We thank the reviewer for pointing out this. We have corrected the figure number in the revised manuscript on page 12.

10. Regarding the ARPES measurements and the possible match with DFT simulations performed for thin 1T CrTe₂ flakes, I have several questions. The authors write "The observed metallic band further confirms the trigonal phase of the CrTe₂ thin films". However, Fig. S9 for instance shows that both the 1T and the 2H phases are predicted

metallic (at least in the bulk form). Actually, I find it very difficult to conclude that the observed electronic band structure resembles the one predicted based on DFT calculations. The bands are actually not nicely resolved, and there are many features that do not really look alike what is calculated (Fig. 4a/b). What about a calculation for the 2H phase of a thin CrTe₂ film (not for bulk, like in Fig. S9)? The number of layers seems an important parameter, and systematic calculations for different layer number might be relevant here. By the way, I might have missed the information, but I did not figure out what the layer number is for Fig. 4a.

All this needs to be clarified and maybe discussed with more care.

Reply: Thanks for the helpful suggestions. Both 1T and 2H phases are indeed predicted to be metallic. We agree that the observed metallic band can't provide the evidence for the crystal structure. We have corrected this statement in the revised manuscript.

For the quality of measured electronic band structure of CrTe₂, actually it is comparable with that of TMDCs (*e.g.* VSe₂¹⁸ and WTe₂¹⁹) and other 2D ferromagnets (*e.g.* Fe₃GeTe₂²⁰ and CrGeTe₃²¹). In order to distinguish the crystal structure more clearly, we put the calculated 1T (left) and 2H (right) bands on the top of the experimental one, as shown in Fig. R4. The ARPES-intensity plots and calculated band structure for 1T-CrTe₂ share most common features. For example, the hole-like bands cross E_F around the Γ point (marked by the green arrow), and a relatively flat Cr 3d orbital band locates at $E_B \sim 1$ eV. The other hole pockets (marked by the purple arrow) are not observed due to the distinct photon-energy responses of majority and minority bands. In Fig. R4b, many obvious differences are displayed in the bands of 2H phase, such as the flat hole-like bands near Γ (marked by the green arrow) and a “M” shape flat band with top at $E_B \sim 0.6$ eV. These characteristic bands establish the intrinsic differences between the electronic states of the 1T and 2H phases, and further confirm the trigonal phase of CrTe₂ thin films.

Fig. R4 Comparison among experimental band dispersion and DFT calculations. Calculated band structure obtained from the first-principles band calculations for (a) 1T and (b) 2H phases compared with the ARPES-intensity plot of 7 ML CrTe₂.

The comparison of theoretical calculation for the bilayer thin film and bulk CrTe₂ with 2H phase is shown below. The main features, including flat hole-like bands near Γ point and a “M” shape flat band, remain unchanged when the thickness changes from bulk into bilayer. Accordingly, the variation of thickness cannot account for the disagreement between the experimental band dispersion and the calculated 2H phase.

Fig. R5 Calculated band structures of 2H-CrTe₂. Comparison of the theoretical calculated electronic structure for and (a) bilayer and (b) bulk 2H-CrTe₂ along the high symmetry direction M- Γ -K.

The sample in Fig. 5a is 7 ML CrTe₂. We have added this important information in the revised manuscript. To understand the thickness-dependent electronic structure, we carried out first-principles calculations of 1T-CrTe₂ with different thicknesses. As shown in Fig. R6, there is an excellent agreement between our experiment and theory. In particular, the hole-like band near E_F and a relatively flat Cr 3d orbital band are similar to that of calculated 1T-CrTe₂ with the inclusion of spin polarization. For the 1 ML film, the two parabolic hole pockets are well reproduced by the majority spin projections of the bands, which highlights the FM nature. These results demonstrate that the epitaxial 1T structure and ferromagnetism have been established since 1 ML deposition.

Fig. R6 Band structures of CrTe₂ ultrathin films. (a) 1 ML. (b) 2 ML. (c) 3 ML. (d) 5 ML. (e) 7 ML. All the spectra were taken along the high symmetry direction M- Γ -M. Upper panels: band structures of 1T phase from first-principles calculations; Middle panels: calculated minority (red) and majority (blue) spin projections of the bands, respectively; Lower panels: ARPES intensity maps.

Revision: In the revised manuscript, we have discussed the comparison among ARPES band and DFT calculations of 1T and 2H phases. The relationship between electronic structure against thickness variation has also been included in the new version of the paper on page 13 and 14.

“The electronic structure of 2H-CrTe₂ was also calculated, as presented in Supplementary Fig. 15. A lot of differences are apparent in the bands of 2H phase, such as the flat hole-like bands near Γ and a “M” shape flat band with top at binding energy (E_B) ~ 0.6 eV. On the contrary, the ARPES-intensity plots and calculated band structure for 1T-CrTe₂ share most common features. For example, the hole-like bands crossing the E_F around Γ point and a relatively flat Cr 3d orbital band at $E_B \sim 1$ eV. These characteristic bands establish the intrinsic differences between the electronic states of the 1T and 2H phases, and further confirm the trigonal phase of CrTe₂ thin films.”

“To understand the thickness-dependent electronic structure, we carried out first-principles calculations of 1T-CrTe₂ with different thicknesses. As shown in Supplementary Fig. 16, there is an excellent agreement between our experiment and theory. In particular, the hole-like band near E_F and a relatively flat Cr 3d orbital band are similar to that of calculated 1T-CrTe₂ with the inclusion of spin polarization. For the 1 ML film, the two parabolic hole pockets are well reproduced by the majority spin projections of the bands, which highlights the FM nature. These results demonstrate that the epitaxial 1T structure and ferromagnetism have been established since 1 ML deposition.”

11. Too few details are given about how the DFT calculations and electronic band structure was simulated.

Revision: Thank you for the comment. Following the suggestion, detailed information about the DFT calculations has been added in Methods on page 16 accordingly.

“First-principles calculations with DFT were performed as implemented in the Vienna ab Initio Simulation Package (VASP) package. We used the Perdew-Burke-Ernzerhof (PBE) form of the exchange correlation functional. All the calculations were performed with a plane-wave cut-off energy of 300 eV on the 11 × 11 × 1 Monkhorst–Pack k-point mesh. The super cell includes CrTe₂ layers with varying thicknesses and a vacuum layer of about 20 Å, in order to avoid interactions between the neighboring slabs. CrTe₂ with an in-plane lattice constant of 3.81 Å was used. The atomic positions and the out-of-plane lattice constant were optimized by the conjugate gradient method. Calculations of the band structures were done including SOC.”

Reviewer #2 (Remarks to the Author):

Comments:

Two-dimensional van der Waals magnet is a new class of materials, being recognized as a unique platform for studying the two dimensional magnetism and also an important building block of van der Waals interfaces for novel spintronic functionalities.

In this work, authors report the epitaxial growth of a new 2D van der Waals magnet CrTe₂ and its magnetic/electronic properties. The results (High transition temperature, perpendicular magnetic anisotropy, and layer dependent magnetic/electronic properties) are clearly indicates the potential of this material.

I feel that results are worth publishing in Nature Communications. However, there are several unclear points and I think the authors should further clarify them.

Reply: First of all, we are grateful to the reviewer for his/her positive comments and the fact that he/she thinks our manuscript are worth publishing in Nature Communications. We also thank the reviewer for the detailed comments and constructive suggestions, which help us to improve our manuscript. The questions raised by the reviewer are answered point by point as follows.

1. Although authors mentioned about the stoichiometry of grown films and excluded the possibilities of CrTe, Cr₂Te₃, and Cr₅Te₈, it requires more careful discussion to conclude the stoichiometry. It is known that MBE-grown sample can have its unique stoichiometry (partially intercalated one, for example), which is different from bulk and can even show the thickness dependence. In order to unambiguously clarify it, authors should show the cross-sectional TEM image of their samples with several different thicknesses.

Reply: We thank the reviewer for the valuable suggestion. TEM characterizations were performed for CrTe₂ thin films with different thicknesses, as exhibited in Fig. R7. The high-resolution TEM shows the $\sqrt{3}a \times a$ arrangement, indicating that the as-grown thin films correspond to the 1T phase with an octahedral (*O_h*) symmetry. The corresponding FFT pattern further demonstrates that CrTe₂ thin films match its supposed 1T structure very well.

Fig. R7 Atomic-resolution HAADF-STEM images of CrTe₂ thin films with different thicknesses. The inset atomic model demonstrates the lattice structure of 1T-CrTe₂ in cross-sectional view.

Note that Cr atoms are fairly light compared with Te, leading to a quite low intensity contribution in TEM images^{22,23}. In order to further give an accurate stoichiometry of MBE-grown chromium chalcogenides, extra STM characterizations were carried out. As shown in Fig. R8, the atomically resolved STM images demonstrate the layer-by-layer growth mode and homogeneously well-structured CrTe₂ thin films. The layered surface morphology with a uniform step height (~0.61 nm) suggests that the films are in a single phase without intercalated ones.

Fig. R8 STM images of CrTe₂ thin films with different thicknesses. a-c) Surface morphology of uniform monolayer (a), 3 ML (b) and 15 ML (c) CrTe₂ thin films. d-f) Atomically resolved STM images of the corresponding as-grown CrTe₂ thin films with a hexagonal structure in (a-c).

Revision: According to reviewer’s suggestions, we have included the HAADF-STEM images and atomically resolved STM of CrTe₂ thin films with different thicknesses in the supplementary materials (Fig. S1, S2) and the relative discussion in the main text on page 6.

“STM measurements carried out on several CrTe₂ thin films with different thicknesses (mono- to few-layer) show similar terraces, indicating the layer-by-layer growth mode and homogeneously well-structured thin films (Supplementary Fig. 1).”

“The atomic-resolution high-angle annular dark-field scanning transmission electron microscopy (HAADF-STEM) images show the $\sqrt{3}a \times a$ arrangement, revealing that

CrTe₂ thin films correspond to the 1T phase with an octahedral (O_h) symmetry (Supplementary Fig. 2). Both TEM and STM characterizations manifest our as-grown films are stoichiometric CrTe₂.”

2. I am confused about the valence number of Cr. Considering the stoichiometry of CrTe₂, it seems that Cr⁴⁺ (and Te²⁻) is reasonable. Why authors can conclude Cr³⁺?

Reply: As the reviewer notes, within an ionic model, Cr⁴⁺ is the nominal valence. This is based on Te²⁻. However, Te²⁻ is a large ion (Shannon crystal radius, 2.07 Å) and Te has relatively low electronegativity (Pauling electronegativity, 2.1), both of which make the ionic model questionable. In fact, the band structure shows evidence for Te-Te bonding in the form of a dispersive Te derived band that crosses the Fermi level in bulk CrTe₂. This leads to a partial rather than full occupation of the Te *p* states, and therefore a Cr valence close to Cr³⁺. This is confirmed by the calculated moments²⁴, which are consistent with experiment and correspond closely to Cr³⁺. Moreover, the observed XAS spectral line shape of Cr in CrTe₂ is in line with that of spinel Cu(Cr,Ti)₂Se₄ polycrystals with Cr³⁺ cations on O_h sites²⁵, providing a direct spectroscopic fingerprint of 1T-type CrTe₂ with predominately Cr³⁺ cations.

It also should be noted that there is a connection between valence and crystal structure. It is common that TMDs (e.g. MoSe₂, MoTe₂, etc.) have several phases (e.g. 1T, 2H, 1T', etc.) that determine the physical properties²⁶⁻²⁸. As exhibited in Fig. R9, in 1T phase with O_h coordination, the 3*d* degenerate orbitals of Cr ions are split into two sets of *e_g* and *t_{2g}* states, with an energy separation of nominal 10 *Dq*. The *t_{2g}* states are at lower energy orbitals than the *e_g* states. In this case, the Cr³⁺ (*d³*) configuration with half-filled *t_{2g}* states causes the reduction of free energy²⁸, which increases the stability of the 1T phase. The result is in good agreement with the theoretical calculations of 3 μ_B /Cr atom made by Fumega *et al.*²⁴. Therefore, 1T-type CrTe₂ is conducting and magnetic. For 2H structure, *d_{z²}* orbital has a lower energy compared with *d_{x²-y²}*, *d_{xy}*, *d_{xz}*, *d_{yz}*. Under this circumstances, Cr⁴⁺ (*d²*) is lower in energy since *d_{z²}* orbital is full-filled²⁸. Therefore, 2H-type CrTe₂ is insulating and non-magnetic.

In this work, according to STM, XRD, XAS and ARPES characterizations, CrTe₂ thin films belong to 1T phase, which energetically favors Cr³⁺. The measured FM response of CrTe₂ thin films also indicates the O_h coordination.

Fig. R9 Schematic illustrations of the *d*-orbital splitting of CrTe₂ with 1T phase in an octahedral crystal field and 2H phase in a trigonal crystal field.

Revision: According to reviewer's comment, we have added the discussion of the valence number of Cr in the main text on page 9.

“Due to the O_h coordination, the 3d orbitals of Cr split into e_g and t_{2g} states with energy separation of nominal 10 Dq. The t_{2g} states are at lower energy orbitals than the e_g states. In this case, the Cr³⁺ (d³) configuration with half-filled t_{2g} states causes the reduction of free energy, in good agreement with the theoretical calculations of 3 μ_B/Cr atom made by Fumega et al.²⁴. The observed XAS spectral line shape is in line with that of spinel Cu(Cr,Ti)₂Se₄ polycrystals with trivalent Cr cations on O_h sites²⁵, further providing a spectroscopic fingerprint of 1T-type CrTe₂ with predominately Cr³⁺ cations.”

3. Related with the above comment, can Cr-Te bond be regarded as covalent bond or ionic bond?

Reply: As mentioned above, CrTe₂ thin films have the 1T phase, so Cr³⁺ (d³) is energetically more favorable. In this case, Cr-Te bond could be regarded as primarily ionic. The Te *p_z* wave functions from two adjacent interfacial Te sublayers overlap at the interlayer region, which is similar to the case of CrSe₂ with magnetic moment of ~3 μ_B/Cr²⁹. According to the DFT calculations based on a modified Hubbard model made by Wang et al.²⁹, both CrTe₂ and CrSe₂ contain an interlayer wave-function overlapped region, which could be effectively considered as an area accumulating appreciable shared charge from the two interfacial Se/Te sublayers. Essentially, approximately three electrons are removed from the Cr atoms to form Cr³⁺. These electrons are distributed over the Te. The bonding also includes some Te-Te bonding as mentioned above.

Revision: In the revised manuscript, we have given an explanation of Cr-Te ionic bond in the main text on page 9.

*“Approximately three electrons are removed from the Cr atoms, and distributed over the Te. In this case, Cr-Te bond could be regarded as primarily ionic. The Te *p_z* wave functions from two adjacent interfacial Te sublayers overlap at the interlayer region and form Te-Te bonding, which is similar to the case of CrSe₂²⁹.”*

4. I think it is impossible to determine the space group from out-of-plane XRD (line 133-136 in page 6).

Reply: We thank the reviewer for pointing it out. We could only tell the lattice constant (*c* = 6.13 Å) from out-of-plane XRD results, but not the space group.

Revision: We revised the relative description in the main text on page 6.

*“The diffraction pattern with perpendicular constant *c* = 6.13 Å is matched to the (001) crystal planes of the standard 1T-type hexagonal structure (*a* = 3.79 Å, *c* = 5.94 Å), rather than 2H phase (*a* = 3.49 Å, *c* = 13.64 Å). With STM, TEM and XRD*

characterizations, the formation of CrTe₂ films with 1T phase and their single-crystalline nature has been confirmed.”

5. For monolayer samples, authors only show the ARPES data. They should show the magnetization and XMCD data for monolayer sample. Because it seems that transition temperature of 3 ML samples is around 250 K (Fig. 2a) and that of monolayer might be lower than it, the word “monolayer” in the title is misleading.

Reply: We thank the reviewer for this constructive suggestion. Indeed, magnetic characterization for the monolayer sample is meaningful. However, it is quite challenging to accurately detect magnetization in such ultrathin films by SQUID, since the magnetic signal of 1 ML CrTe₂ is too weak compared with the overwhelmingly larger background signal from the substrate and beyond the resolution of SQUID. Therefore, we did element-specific XMCD characterization with a much higher sensitivity of 1 ML CrTe₂ film.

Figure R10 shows the XAS and XMCD spectra of 1 ML CrTe₂ film at Cr $L_{2,3}$ edges taken at different temperatures under out-of-plane magnetic field of 1 T. There is a clear difference in the XAS spectra between left-handed circularly polarized and right-handed circularly polarized setups, indicating the existence of the XMCD signals. Although the dichroism is small compared with 7 ML sample, the clear XMCD signals ($\sigma^+ - \sigma^-$ is nonzero) appear near the absorption peaks. It suggests that the intrinsic ferromagnetism of 1 ML CrTe₂ film originates from the spin polarization of Cr 3d electrons. Accurate calculation of the magnetic moment remains a challenge since the contribution of Te capping layer to the XAS spectra is so large for 1 ML sample. The XMCD percentage increases with decreasing temperature, in line with the typical FM behavior. The nonzero XMCD percentage persists when temperature approaches 200 K and disappears at 250 K, indicating that 1 ML CrTe₂ has a T_C of ~200 K.

We agree with the reviewer that the word “monolayer” in the title may cause misunderstanding. In the revised manuscript the title has been modified to: “**Room-temperature intrinsic ferromagnetism in epitaxially grown CrTe₂ atomic-layer films.**”

Fig. R10 Temperature dependent XMCD characterization of 1 ML CrTe₂ film. (a) Typical pairs of XAS and XMCD spectra of 1 ML CrTe₂ thin film at various temperatures, where dichroism at Cr L₃ edge can be traced to 200 K (spectra at different temperatures are offset for clarify). (b) The partial enlarged XAS spectra of 1 ML CrTe₂, where the difference between left- and right-circularly polarized XAS is evident. (c) XMCD percentage versus temperature derived from (a) with the trend guiding lines.

Revision: Following reviewer’s requests, we have included the XMCD study of 1 ML CrTe₂ film in Fig. 4, and also modified the title of the manuscript accordingly.

“The magnetic response of 1 ML CrTe₂ film is worth exploring. It is difficult to detect magnetization in such ultrathin films by SQUID, since the magnetic signal of 1 ML CrTe₂ is too weak compared with an overwhelmingly larger background signal from the substrate and beyond the resolution of SQUID. Therefore, we did element-specific XMCD characterization of 1 ML CrTe₂ film (Fig. 4a). There is a clear difference in the XAS spectra between left- and right-handed circularly polarized setups (Fig. 4b). Although the dichroism is small compared with 7 ML sample, the clear XMCD signals appear near the absorption peaks. It suggests that the intrinsic ferromagnetism of 1 ML CrTe₂ film originates from the spin polarization of Cr 3d electrons. Accurate calculation of the magnetic moment remains a challenge since the contribution of Te capping layer to the XAS spectra is so large for 1 ML sample. The XMCD percentage increases with decreasing temperature (Fig. 4c), in line with the typical FM behavior. The nonzero XMCD percentage persists when temperature approaches 200 K and disappears at 250 K, indicating that 1 ML CrTe₂ has a T_C of ~200 K.”

6. They only show the ARPES data at 107 K. More detailed discussion about the temperature dependence of ARPES data (temperature dependent spin splitting value) is required.

Reply: As suggested by reviewer, to understand the interplay between the band structure and magnetic properties, we carried out the ARPES measurements of 7 ML CrTe₂ at 107 K and 300 K, respectively. As shown in Fig. R11, the typical band dispersion hardly changes except for the thermal broadening with increasing temperature.

The energy splitting size could be obtained from the temperature dependent energy distribution curves (EDCs) of the second derivative data. For clarity, the temperature evolution of the EDCs at $k = -0.36 \text{ \AA}^{-1}$ is summarized in Fig. R11e. The fitted peak-to-peak splitting of majority and minority bands at 300 K shows an obvious decreasing trend compared with 107 K, corresponding to the weaker ferromagnetism at higher temperature.

Fig. R11 Spin splitting at different temperatures. a-d) Temperature evolution of second-derivative valence structure near the E_F taken by He I α (21.2 eV) (a,c), He II α photons (40.8 eV) (b,d) along the high symmetry direction M- Γ -M. e) EDCs of panel (a-d) at $k = -0.36 \text{ \AA}^{-1}$. Peak positions of hole bands are indicated by the arrows.

Revision: We have included the discussion of temperature dependent splitting of hole pockets in the main text on page 13.

“The energy splitting value could be obtained from the temperature dependent energy distribution curves (EDCs) of the second derivative data (Supplementary Fig. 14). The fitted peak-to-peak splitting of majority and minority bands at 300 K shows an obvious decreasing trend compared with 107 K, corresponding to the weaker ferromagnetism at higher temperature.”

7. What is the mechanism of perpendicular magnetic anisotropy? Is it consistent with the bulk CrTe₂?

Reply: The origin of magnetic anisotropy generally originates from three mechanisms, shape anisotropy^{10,11}, magneto-crystalline anisotropy¹² and anisotropy of exchange interactions¹¹. The shape anisotropy is due to the magnetostatic or dipole interactions, which lead to a preferential in-plane anisotropy for thin films¹¹. The PMA in CrTe₂ thin films is contrary to it. Thereby, the possibility of shape anisotropy can be ruled out.

The anisotropy of exchange interactions can be verified by performing DFT calculations of exchange interactions using the frozen magnon approach¹². Fujisawa *et al.* found that the ferromagnetic (FM) Cr-Cr intra-sublattice exchange interaction dominates in CrTe₂ thin films, and the total energy minima for CrTe₂ occurs at perpendicular direction¹². In this vein, the magnetic moments of CrTe₂ thin films are driven to perpendicular direction due to the anisotropy of exchange interactions.

CrTe₂ few-layer films show a strong PMA. It is contradictory to the bulk CrTe₂ reported by Freitas *et al.*⁸, which possesses in-plane easy axis. The thickness dependent magnetic anisotropy suggests that the lowered symmetry at the interface plays an important role in determining the PMA in CrTe₂ thin films. As the magnetic film thickness approaches a few nm, interfacial magnetism and inversion symmetry breaking give rise to intriguing phenomena such as PMA¹³. This is a consequence of spin-orbit interactions, *i.e.*, magneto-crystalline anisotropy, that apparently have a stronger effect in the more anisotropic film limit^{10,14,15}. Both experimental results and DFT calculations indicate that, the magnetic moments of CrTe₂ thin films are driven to the perpendicular direction, due to the magneto-crystalline anisotropy and the anisotropy of exchange interactions.

Revision: According to reviewer's suggestions, we have included the discussion of PMA in CrTe₂ films on page 8.

*"The strong PMA in CrTe₂ few-layer films is different from bulk CrTe₂⁸ and exfoliated flakes (thicker than 10 nm)¹⁶ which have an in-plane easy axis. Here, the thickness dependent magnetic anisotropy suggests that the reduced symmetry at the interface plays an important role in determining the PMA in CrTe₂ thin films¹⁵. As the magnetic film thickness approaches a few nm, the interfacial magnetism and inversion symmetry breaking give rise to the PMA¹³. This is a consequence of spin-orbit interactions, *i.e.*, magneto-crystalline anisotropy, that apparently have a stronger effect in the more anisotropic film limit^{10,14,15}. In addition, based on density functional theory (DFT) calculations, it has been found that the FM Cr-Cr intrasublattice exchange interactions dominate in CrTe₂ thin films, and the total energy minima is at perpendicular direction¹². In general, the magnetic moments of CrTe₂ thin films are aligned in the perpendicular direction, due to the magneto-crystalline anisotropy and the anisotropy of exchange interactions."*

8. In Fig. 2a, it is written that magnetic field is applied in-plane. It is correct? If so, it is

better to replace it with that under the out-of-plane magnetic field.

Reply: Yes, the magnetic field is applied in-plane in Fig. 2a. Thanks for the nice suggestion. We have replaced the in-plane temperature dependent magnetization (M - T) curves with the out-of-plane ones, as shown in Fig. R12. The high T_C (~258 K) is preserved with thickness decreasing to 3 ML.

Fig. R12 Temperature dependent magnetization of CrTe₂ thin films under field-cooled mode. The magnetic field is applied out-of-plane with a magnitude of 1000 Oe.

Revision: According to reviewer's suggestions, we have replaced the in-plane M - T curves with out-of-plane ones in the revised manuscript on page 7.

"The temperature dependent magnetization (M - T) curves of CrTe₂ thin films with different thicknesses under out-of-plane magnetic field were measured, as shown in Fig. 2a. It shows a general trend of decreasing with the increase of temperature, demonstrating a FM nature."

9. 15ML data in Fig. 2a shows slight decrease at low temperature? What is the origin of this strange behavior?

Reply: The slight decrease at low temperature for the in-plane measurement could be ascribed to the accelerated process of spin reorientation from the ab plane to the c axis with decreasing temperature at low magnetic field³⁰. Based on the fact that CrTe₂ film has a strong PMA ($K_u = 5.63 \times 10^6$ erg/cm³ for 7 ML CrTe₂), spin intends to align along the perpendicular direction at low temperature.

We did control experiment about the temperature dependence of out-of-plane magnetization with magnetic field of 0.1 T, as shown in Fig. R13. Note that out-of-plane direction is an easy axis. Contradictory to the in-plane one, the out-of-plane magnetization increases with decreasing temperature, indicating the enhanced ferromagnetism at low temperature. Therefore, the slight decrease of in-plane magnetization at low temperature is resulted from the spin reorientation from the ab plane to the c axis, due to the weak in-plane magnetic field (0.1 T).

Fig. R13 Zero-field and field cooled temperature dependence of the magnetization of 7 ML CrTe₂ with an applied in-plane and out-of-plane magnetic field of 0.1 T.

Moreover, we also did control experiment about the temperature dependence of in-plane magnetization with a higher magnetic field of 0.5 T. As shown in Fig. R14, the in-plane magnetization at low temperature does not decrease at a high magnetic field, further indicating that the slight decrease of in-plane magnetization at low temperature is due to the spin reorientation from in-plane to the out-of-plane direction.

Fig. R14 Field cooled temperature dependence of the magnetization of 7 ML CrTe₂ thin film with an applied in-plane magnetic field of 0.1 T and 0.5 T, respectively.

Revision: According to reviewer’s advices, we have replaced the in-plane M - T curves with out-of-plane ones in the revised manuscript on page 7, and discussed the origin of slight decrease of in-plane M - T curves at low temperature in Fig. S3.

“The out-of-plane magnetization increases with decreasing temperature, indicating the enhanced ferromagnetism at low temperature. On the contrary, the slight decrease of in-plane magnetization at low temperature is resulted from the accelerated process of spin reorientation from the ab plane to the c axis, due to the weak in-plane magnetic field.”

10. Authors claim that orbital magnetization is almost negligible. If so, why magnetic anisotropy appears?

Reply: Although the orbital moments are small due to the crystal field scheme, they are apparently sufficient to yield a magneto-crystalline anisotropy and the perpendicular orientation of the moments that underlies the FM order in this 2D system. This has also been found in other systems such as Cr-doped Bi_2Se_3 ³¹ and CrI_3 ($\sim 3 \mu_{\text{B}}/\text{Cr}$)¹¹. At the same time, the anisotropy of magnetic exchange interactions also contribute to the PMA¹².

Revision: According to reviewer's comments, we have discussed the relationship between the orbital magnetization and magnetic anisotropy in the revised manuscript on page 10.

“Although the orbital moments are small due to the crystal field scheme, they are apparently sufficient to yield a magneto-crystalline anisotropy and the perpendicular orientation of the moments that underlies the ferromagnetic order in this 2D system, similar to that of Cr-doped Bi_2Se_3 ³¹ and CrI_3 ¹¹.”

Reviewer #3 (Remarks to the Author):

Comments:

Dear Authors,

In this manuscript, the authors report the observation of room temperature ferromagnetism in CrTe₂ thin films. The manuscript is organized with interesting experimental results about structural characterization (Fig. 1 and S1), magnetization (Fig. 2), anomalous Hall effect (Fig. S2), circular dichroism (Fig. 3 and S3), and ARPES with calculation (Fig. 4 and S4-S9). I totally agree that the successful fabrication of high quality CrTe₂ ultrathin films is quite important advancement. By applying the interface formation, the spintronic function can be expected to work at room temperature. However, in previous studies, the room temperature ferromagnetism has been observed in bulk and exfoliated thin flakes of CrTe₂. Of course, XMCD and ARPES are also important experimental results to understand the electronic structures of the CrTe₂ films. Considering these points, for my sense, it is difficult to find new significant achievements in the preset manuscript.

I provide some comments to improve the manuscript.

Reply: We appreciate the reviewer for pointing out the importance of the successful fabrication of high quality CrTe₂ ultrathin films for spintronics. We also understand the inquires of the reviewer about the comparison with the previous work on bulk and exfoliated thin flakes of CrTe₂ (Note: The magnetic properties of the exfoliated thin flakes as reported in refs^{9,16} are essentially similar to that of the bulk with in-plane anisotropy, but with enhanced H_C). Here, we would like to highlight the significance and novelty of our work, especially after substantial new results added in the revised manuscript following the suggestions from all three reviewers..

- 1) The successful synthesis of CrTe₂ **epitaxial films by MBE with controllable size and thicknesses**. What is currently stunting the progress in the research of 2D magnetic materials is the lack of stoichiometric 2D materials with intrinsic ferromagnetism and compatibility with large scale solid state device applications, where the MBE growth is critical..
- 2) The scaling of the CrTe₂ thickness down to **monolayer**. Remarkably, a high T_C of **~200 K** is still retained with the film thickness down to **monolayer**, suggesting the robust ferromagnetism in the epitaxially grown CrTe₂ thin films. Actually it is much higher than those recently reported ML 2D ferromagnets, such as Fe₃GeTe₂ (20 K), CrI₃ (45 K).
- 3) The experimental demonstration of **strong PMA** and large coercivity in CrTe₂ thin films. It is contradictory to the previous studies of CrTe₂ bulk and exfoliated flakes with in-plane easy axis. The PMA is not only an intriguing fundamental issue, but also important for applications such as magnetic tunneling junctions (MTJs). MTJs with PMA require a smaller switching current and have a faster reversal speed for magnetization switching than one with in-plane anisotropy.

Furthermore, materials with large coercivity and PMA represent the mainstay of data storage media, e.g. MRAM, owing to their ability to retain a permanent and stable magnetization state.

- 4) The **first** ARPES observation of **magnetic splitting band structure** of 2D magnets, revealing the unique interplay between macroscopic ferromagnetism and atomic electronic structure in CrTe₂. It opens a new pathway to determine the intrinsic ferromagnetism in 2D magnets.

Revision: Following the reviewer's comments, we have improved the discussion about the significance and novelty of this work along with the new results added to the revised manuscript page 5 and 11.

“Very recently, a paper reported the observation of above room-temperature ferromagnetism in the exfoliated thin flakes of CrTe₂ (10 nm, or ~17 ML)¹⁶. Their properties were found to be rather similar to that of the bulk with in-plane anisotropy, but with enhanced H_c compared with its bulk counterpart. However, the magnetic response (e.g., T_C and PMA) of CrTe₂ epitaxial thin films with the thickness down to monolayer limit has not been explored so far.”

“In order to investigate the dimensionality effect of the ferromagnetism in CrTe₂ stemming from thermal fluctuation, we plot the thickness dependent T_C (Fig. 4d). The T_C of CrTe₂ decreases slightly with reducing the film thickness compared with other commonly investigated 2D magnets, such as Cr₂Ge₂Te₆² and Fe₃GeTe₂⁷ (Supplementary Fig. 8), demonstrating the robustness of ferromagnetism in the epitaxially grown CrTe₂ thin films.”

In the following, we address the specific points raised by the reviewer:

1, appropriate definition should be used.

The definition in this manuscript should be carefully improved, for example “intrinsic ferromagnetism” in the title and “intrinsic 2D ferromagnetism” in the abstract, are described but no discussion is provided in main text. What does “intrinsic” mean? How did the authors conclude “2D” ferromagnetism? The meaning of “2D magnet” is a 2D material possessing magnetism. It is generally accepted. However, the meaning of “Intrinsic 2D ferromagnetism” is completely different from that of 2D magnet. The dimensionality of the ferromagnetism has to be examined.

Reply: The discovery of intrinsic ferromagnetism in 2D materials is vital for understanding the spin behavior in low dimensions. Using “intrinsic ferromagnetism,” we intend to limit the discussion to ferromagnetism occurring in a stoichiometric compound or heterostructure of stoichiometric compounds, as opposed to defect or dopant induced (“dilute”) ferromagnetism³². The term, “2D ferromagnetism,” refers to ferromagnetism in 2D materials, which can be stably isolated in atomically thin layers. We agree with the reviewer's opinion that the statement of “intrinsic 2D

ferromagnetism” might be not rigorous. According to reviewer’s suggestions, we have changed the statement of “intrinsic 2D ferromagnetism” into “intrinsic ferromagnetism in 2D CrTe₂ films”.

In order to investigate the dimensionality effect of the ferromagnetism in CrTe₂, we plot the thickness dependent T_C in Fig. R15. The normalized T_C of CrTe₂ decreases slightly with reducing film thickness, displaying a weak dimensionality effect, compared with that of Fe₃GeTe₂ for example.

Fig. R15 T_C (normalized to bulk T_C for the particular material) as a function of the number of layers. Data for Fe₃GeTe₂ adapted from ref. ⁷; for Cr₂Ge₂Te₆, ref. ².

Revision: According to reviewer’s suggestions, we have changed the statement of “intrinsic 2D ferromagnetism” into “intrinsic ferromagnetism in 2D CrTe₂ films”. We have also included a brief discussion of “intrinsic ferromagnetism” and the dimensionality of the ferromagnetism in the revised manuscript on page 3 and 11.

“Opposed to defect or dopant induced magnetism, the ferromagnetism occurring in a stoichiometric compound is defined as intrinsic ferromagnetism.”

“In order to investigate the dimensionality effect of the ferromagnetism in CrTe₂ stemming from thermal fluctuation, we plot the thickness dependent T_C (Fig. 4d). The T_C of CrTe₂ decreases slightly with reducing the film thickness compared with other commonly investigated 2D magnets, such as Cr₂Ge₂Te₆² and Fe₃GeTe₂⁷ (Supplementary Fig. 8), demonstrating the robustness of ferromagnetism in the epitaxially grown CrTe₂ thin films.”

If the films have less defect concentration like bulk single crystals, the observed ferromagnetism is supported by similar origin in the bulk crystals. It is great that the authors explain the basis for what is intrinsic and extrinsic magnetism.

Reply: In magnetism, there is a fundamental distinction between intrinsic and extrinsic

properties. Several approaches have been proposed to extrinsically induce long-range magnetic order into 2D materials such as defect engineering, absorption of magnetic ions, or proximity effect. In defect engineering, the simplest strategy for realizing magnetism in non-magnetic materials is to create unpaired electrons by modifying its electronic structure via vacancies, atoms, grain boundaries, or edges. In absorption technique, magnetic ions are absorbed inside 2D vdW materials where magnetic order emerges when the magnetic ions experience exchange coupling. On the other hand, in magnetic proximity effect, 2D vdW materials experience exchange coupling when placed in contact with FM insulating substrates.

However, little attention has been drawn to investigate the intrinsic ferromagnetism in 2D materials because the long-range magnetic order in 2D systems could be destroyed by thermal fluctuations. For example, the Mermin-Wagner theorem indicates that long-range magnetic order is impossible at any finite temperature in an isotropic 2D spin system. To suppress the thermal fluctuations and achieve long range intrinsic ferromagnetism in 2D materials, magnetic anisotropy is necessary. Based on various forms of magnetic anisotropies, recent groundbreaking experiments of intrinsic ferromagnetism in 2D materials have been realized in $\text{Cr}_2\text{Ge}_2\text{Te}_6$ ² and CrI_3 ¹.

In this work, by the term of “intrinsic ferromagnetism,” we intend to limit the discussion to ferromagnetism occurring in a stoichiometric compound or heterostructure of stoichiometric compounds, as opposed to defect or dopant induced (“dilute”) ferromagnetism.

Revision: According to the reviewer’s advice, we have included a definition of “intrinsic ferromagnetism” in the revised manuscript on page 3.

“Opposed to defect or dopant induced magnetism, the ferromagnetism occurring in a stoichiometric compound is defined as intrinsic ferromagnetism.”

Room temperature ferromagnetism is obtained in only 7 and 15 ML films. These thicknesses are comparable to that in the previous studies of exfoliated flakes. Please describe the results with fair comparison. The plot of Curie temperature as a function of thickness is quite meaningful, because the analysis for Curie temperature of each films is not clearly explained.

Reply: We fully understand the reviewer’s comment. As we discussed before, the properties of the exfoliated thin flakes are essentially equivalent to the bulk with in-plane anisotropy according to the result reported by Sun *et al*¹⁶. The magnetization of any exfoliated CrTe_2 flakes thinner than 10 nm hasn’t been explored for the limitation of noise level of Faraday experimental setup and the technical difficulties of exfoliating thinner CrTe_2 . Here, we have achieved the epitaxial growth of CrTe_2 compatible with large-scale solid state device applications, room-temperature ferromagnetism down to 7 ML with PMA and ferromagnetism with T_C of 200K in the monolayer films. A fair comparison has been added to the revised manuscript as we discussed in the very first

part.

We agree with the reviewer that the plot of T_C as a function of thickness is quite meaningful. Indeed, a method to determine T_C is to find the temperature at which the remanent magnetization signal goes to zero. In fact, it is this criterion that distinguishes a ferromagnet from a paramagnet. This is the approach that Gong *et al.*² took in searching for ferromagnetism in atomically-thin $\text{Cr}_2\text{Ge}_2\text{Te}_6$. The out-of-plane M - T curves and plot of T_C as a function of thickness is shown in Fig. R16. By XMCD characterization, the high T_C (~ 200 K) is demonstrated to preserve with thickness decreasing to monolayer. The T_C of CrTe_2 decreases slightly with reducing the film thickness compared with other commonly investigated 2D magnets, such as $\text{Cr}_2\text{Ge}_2\text{Te}_6^2$ and $\text{Fe}_3\text{GeTe}_2^7$, demonstrating the robustness of ferromagnetism in the epitaxially grown CrTe_2 thin films.

Fig. R16 M - T curves of CrTe_2 thin films with different thicknesses. (a) Temperature dependent magnetization of CrTe_2 thin films under field-cooled mode. The magnetic field is applied out-of-plane with a magnitude of 1000 Oe. (b) Thickness dependence of the T_C .

Revision: According to reviewer’s suggestions, we have included a fair comparison of this work and previous studies and a plot of thickness dependent T_C in the main text on page 5 and 11.

“Very recently, a paper reported the observation of above room-temperature ferromagnetism in the exfoliated thin flakes of CrTe_2 (10 nm, or ~ 17 ML)¹⁶. Their properties were found to be rather similar to that of the bulk with in-plane anisotropy, but with enhanced H_c compared with its bulk counterpart. However, the magnetic response (e.g., T_C and PMA) of CrTe_2 epitaxial thin films with the thickness down to monolayer limit has not been explored so far.”

“In order to investigate the dimensionality effect of the ferromagnetism in CrTe_2 stemming from thermal fluctuation, we plot the thickness dependent T_C (Fig. 4d). The T_C of CrTe_2 decreases slightly with reducing the film thickness compared with other commonly investigated 2D magnets, such as $\text{Cr}_2\text{Ge}_2\text{Te}_6^2$ and $\text{Fe}_3\text{GeTe}_2^7$ (Supplementary Fig. 8), demonstrating the robustness of ferromagnetism in the epitaxially grown CrTe_2

thin films.”

2, magnetic anisotropy should be discussed.

In this manuscript, the perpendicular magnetic anisotropy (PMA) is claimed. This feature is different from that in bulk and exfoliated flakes. However, the reason is not discussed well. I recommend to discuss the reason why PMA is obtained in the films. Although Fig. 2a shows temperature dependence of in-plane magnetization, temperature dependence of perpendicular magnetization should be also plotted if the authors try to claim the perpendicular magnetic anisotropy.

Reply: We appreciate the reviewer’s comment and suggestion. We did control experiment about the temperature dependence of out-of-plane and in-plane magnetization with magnetic field of 0.1 T, as shown in Fig. R17. For CrTe₂ thin films, the out-of-plane direction is an easy axis, consistent with experimental result reported by Y. Fujisawa¹².

Fig. R17 Zero-field and field cooled temperature dependence of the magnetization of 7 ML CrTe₂ with an applied in-plane and out-of-plane magnetic field of 0.1 T.

The origin of magnetic anisotropy generally originates from three mechanisms, shape anisotropy^{10,11}, magneto-crystalline anisotropy¹² and anisotropy of exchange interactions¹¹. The shape anisotropy is due to the magnetostatic or dipole interactions, which lead to a preferential in-plane anisotropy for thin films¹¹. The PMA in CrTe₂ thin films is contrary to it. Thereby, the possibility of shape anisotropy can be ruled out.

The anisotropy of exchange interactions can be verified by performing DFT calculations of exchange interactions using the frozen magnon approach¹². Fujisawa *et al.* found that the ferromagnetic (FM) Cr-Cr intra-sublattice exchange interaction dominates in CrTe₂ thin films, and the total energy minima for CrTe₂ occurs at perpendicular direction¹². In this vein, the magnetic moments of CrTe₂ thin films are driven to perpendicular direction due to the anisotropy of exchange interactions.

According to the result reported by Freitas *et al.* and Sun *et al.*, bulk CrTe₂⁸ and

exfoliated flakes^{9,16} show in-plane easy axis. Note that the thickness of 1T-CrTe₂ exfoliated flakes showing in-plane easy axis is larger than 10 nm. Here, the thickness dependent magnetic anisotropy suggests that the lowered symmetry at the interface plays an important role in determining the PMA in CrTe₂ thin films. As the magnetic film thickness approaches a few nm, interfacial magnetism and inversion symmetry breaking give rise to intriguing phenomena such as PMA¹³. This is a consequence of spin-orbit interactions, *i.e.*, magneto-crystalline anisotropy, that apparently have a stronger effect in the more anisotropic film limit^{10,14,15}. Both experimental results and DFT calculations indicate that, the magnetic moments of CrTe₂ thin films are driven to the perpendicular direction, due to the magneto-crystalline anisotropy and the anisotropy of exchange interactions.

Revision: According to reviewer's suggestions, we have included the discussion of PMA in CrTe₂ thin films on page 8 and the temperature dependence of perpendicular magnetization in Fig. 2a on page 7.

*“The strong PMA in CrTe₂ few-layer films is different from bulk CrTe₂⁸ and exfoliated flakes (thicker than 10 nm)¹⁶ which have an in-plane easy axis. Here, the thickness dependent magnetic anisotropy suggests that the reduced symmetry at the interface plays an important role in determining the PMA in CrTe₂ thin films¹⁵. As the magnetic film thickness approaches a few nm, the interfacial magnetism and inversion symmetry breaking give rise to the PMA¹³. This is a consequence of spin-orbit interactions, *i.e.*, magneto-crystalline anisotropy, that apparently have a stronger effect in the more anisotropic film limit^{10,14,15}. In addition, based on density functional theory (DFT) calculations, it has been found that the FM Cr-Cr intrasublattice exchange interactions dominate in CrTe₂ thin films, and the total energy minima is at perpendicular direction¹². In general, the magnetic moments of CrTe₂ thin films are aligned in the perpendicular direction, due to the magneto-crystalline anisotropy and the anisotropy of exchange interactions.”*

“The temperature dependent magnetization (M-T) curves of CrTe₂ thin films with different thicknesses under out-of-plane magnetic field were measured, as shown in Fig. 2a. It shows a general trend of decreasing with the increase of temperature, demonstrating a FM nature.”

In Fig. 2a, the small magnetization actually remains at 300 K with weak temperature dependence. Judging from this two-step-like temperature dependence, it is natural that two magnetic origins play a role in this film, in other words, weak dependent component around 300K and strong dependent component at low temperature region. The authors should clarify the two origins in the M-T curve by analysis for usual magnet with critical exponent beta term.

Reply: Thanks for the nice comments. The weak temperature dependence of magnetization near T_C is commonly observed in ferromagnets, including CrGeTe₃²¹, CrSiTe₃³³, Fe₃GeTe₂³⁴, Cr_{1+x}Te₂¹², CrCl₃³, etc. This kind of “tail” phenomenon can be

explained by a positive-feedback mean-field modification of the classical Brillouin magnetization theory³⁵ (Fig. R18). The modified theory incorporates the temperature-dependent quantum-scale hysteretic and mesoscopic domain-scale anhysteretic magnetization processes. Moreover, it includes the effects of demagnetization and exchange fields. It is found that the thermal behavior of the reversible and irreversible segments of the hysteresis loops, as predicted by the theory, is a key to the presence or absence of the “tails.”

Fig. R18 Theoretical temperature-dependence of reduced spontaneous magnetization in Ni compared with measurement³⁵.

Revision: Following the reviewer’s suggestions, we have included the discussion of weak temperature dependence of magnetization near T_C in the revised manuscript on page 7.

“The magnetization exhibits weak temperature dependence near T_C , which is commonly observed in ferromagnets^{2,21,34}. It can be explained by a positive-feedback mean-field modification of the classical Brillouin magnetization theory³⁵.”

The authors claim that the saturation magnetization value of the films is consistent to the value in the bulk. However, it is apparent that the magnetization value does not linearly follow the thickness in Fig. 2a. The magnetization value should be calculated and presented by number of moment for Cr atoms (μ_B/Cr) as discussed in Fig. 3d. Then, the values can be compared to that in bulks and that evaluated by XMCD. If the magnetization values in the films depends on the thickness, the origin should be discussed.

Reply: We thank the reviewer for this advice. The M - T curves in Fig. 2a are measured in-plane under a magnetic field of 0.1 T, which is not high enough to saturate the magnetization. To fairly compare the magnetization of CrTe_2 thin films with different thicknesses, we plot the out-of-plane M - H curves, and present it by number of moment for Cr atoms (μ_B/Cr). As shown in Fig. R19, the magnetic moments of 3 ML and 5 ML CrTe_2 thin films at 10 K are found to be $2.81 \mu_B/\text{Cr}$ and $2.83 \mu_B/\text{Cr}$, respectively. These magnetic moments values are comparable with the theoretical one ($\sim 3 \mu_B/\text{Cr}$)²⁴ and do

not depend on the thickness.

Fig. R19 Out-of-plane magnetic hysteresis loops of 3 ML and 5 ML CrTe₂ thin films.

Revision: Following reviewer’s advices, we have discussed the magnetic moment of 3 ML and 5 ML CrTe₂ thin films in the main text on page 7 and 8.

“In order to clarify the thickness dependence of the magnetic properties, we have measured the field dependent magnetization curves of 3 ML and 5 ML CrTe₂ thin films under out-of-plane and in-plane configuration (Supplementary Fig. 5). The nearly square-shaped FM hysteresis loops under out-of-plane magnetic field suggest the robust FM order with the easy axis perpendicular to the thin films. At 10 K, the magnetic moments of 3 ML and 5 ML CrTe₂ are found to be 2.81 μ_B/Cr and 2.83 μ_B/Cr , respectively, which are comparable with the theoretical value ($\sim 3 \mu_B/\text{Cr}$).”

3, relationship between magnetization and anomalous Hall effect.

Probably the authors already notice, anomalous Hall (AH) effect reflects the perpendicular magnetization. In Fig. S2, the AH resistance for 3 ML film apparently disappears at 250 K, indicating no or rather weak perpendicular magnetization at 250 K. The AHE is measured with 3-layers film so that the M-T curve for 3ML shown in Fig. 2a seems consistent. However, M-H curve in Fig. 2b is measured with 7ML film. To remove such ambiguity, the authors should show all data set to compare Curie temperature and coercive field between AHE and magnetization for 3ML, 5 ML, and 7 ML, if the authors try to claim that the ferromagnetic order can be obtained in ultrathin 3 ML with perpendicular magnetic anisotropy. Perpendicular magnetic anisotropy should be concluded by the discussion and experimental results on the comparison of M-H curves in-plane and out-of-plane. In the present manuscript, the data set of M-H curves for 3 ML and 5 ML is not provided. The authors have to show these experimental evidences.

Reply: We thank the reviewer for this suggestion. The AH resistance of CrTe₂ thin films with thickness of 3 ML, 5 ML and 7 ML is shown in Fig. R20. The AHE behaviors of 3 ML, 5 ML and 7 ML CrTe₂ persist up to 250 K, 250 K and 300 K, respectively. Moreover, the calculated coercive fields by AHE and M-H loops show a good

agreement.

Fig. R20 Electrical transport measurements of CrTe₂ thin films. a-c) The Hall resistance of CrTe₂ thin films with different thicknesses at various temperatures. d-f) Comparison of coercivities measured by AHE and M-H loops, showing a good agreement.

In order to clarify the PMA in ultrathin CrTe₂ thin films, we did field dependent magnetization curves of 3 ML and 5 ML CrTe₂ thin films under out-of-plane and in-plane configuration (Fig. R21). The nearly square-shaped FM hysteresis loops under out-of-plane magnetic field suggest the robust FM order with the easy axis perpendicular to the thin films. The PMA constants ($K_u = \frac{H_K M_S}{2}$) of 3 ML and 5 ML CrTe₂ films at 10 K are determined to be 6.6×10^6 erg/cm³ and 6.5×10^6 erg/cm³, respectively.

Fig. R21 M - H loops of CrTe_2 thin films with different thicknesses. a-d) Temperature dependent magnetic hysteresis loops of 3 ML (a,c) and 5 ML (b,d) CrTe_2 thin films with out-of-plane and in-plane field, indicating a strong out-of-plane magnetic anisotropy.

Revision: Following the reviewer’s comments, we have included the electrical transport measurements and magnetic hysteresis loops of CrTe_2 thin films as supplementary materials (Fig. S5, S6), and discussed them in the revised manuscript on page 7 and 8.

“In order to clarify the thickness dependence of the magnetic properties, we have measured the field dependent magnetization curves of 3 ML and 5 ML CrTe_2 thin films under out-of-plane and in-plane configuration (Supplementary Fig. 5). The nearly square-shaped FM hysteresis loops under out-of-plane magnetic field suggest the robust FM order with the easy axis perpendicular to the thin films. At 10 K, the magnetic moments of 3 ML and 5 ML CrTe_2 are found to be $2.81 \mu_B/\text{Cr}$ and $2.83 \mu_B/\text{Cr}$, respectively, which are comparable with the theoretical value ($\sim 3 \mu_B/\text{Cr}$)²⁴. With in-plane and out-of-plane M - H loops, the PMA constants of 3 ML and 5 ML CrTe_2 films are determined to be $6.6 \times 10^6 \text{ erg/cm}^3$ and $6.5 \times 10^6 \text{ erg/cm}^3$ at 10 K, respectively. The anomalous Hall behaviors of 3 ML, 5 ML and 7 ML CrTe_2 were also investigated, which persist up to 250 K, 250 K and 300 K, respectively (Supplementary Fig. 6). Moreover, the calculated coercive fields by anomalous Hall effect (AHE) and M - H loops show a good agreement.”

In addition, what happen in 2 ML and 1 ML film? The authors describe “indicating a negligible dimensionality effect” in page 5, but the Curie temperature and

magnetization decreases with decreasing thickness. Please reconsider this point. I agree that it is difficult to detect magnetization in such ultrathin films by SQUID but they may detect it by XMCD. If the authors have measured those films, it is great to provide the discussion for 2 ML and 1 ML. By ARPES measurement, electronic bands are well discussed with increasing thickness. Because ferromagnetism is observed above 3 ML, did the authors detect the critical difference between 1, 2 ML and 3 ML in terms of electronic structures? The relationship between electronic structure and magnetism against thickness variation is an interesting topic in such 2D layered magnetic compounds.

Reply: According to reviewer’s constructive suggestions, we have done the element-specific XMCD characterization of 1 ML CrTe₂ film.

Figure R22 shows the XAS and XMCD spectra of 1 ML CrTe₂ film at Cr *L*_{2,3} edges taken at different temperatures under out-of-plane magnetic field of 1 T. There is a clear difference in the XAS spectra between left-handed circularly polarized and right-handed circularly polarized setups, indicating the existence of the XMCD signals. Although the dichroism is small compared with 7 ML sample, the clear XMCD signals ($\sigma^+ - \sigma^-$ is nonzero) appear near the absorption peaks. It suggests that the intrinsic ferromagnetism of 1 ML CrTe₂ film originates from the spin polarization of Cr 3*d* electrons. Accurate calculation of the magnetic moment remains a challenge since the contribution of Te capping layer to the XAS spectra is so large for 1 ML sample. The XMCD percentage increases with decreasing temperature, in line with the typical FM behavior. The nonzero XMCD percentage persists when temperature approaches 200 K and disappears at 250 K, indicating that 1 ML CrTe₂ has a *T*_C of ~200 K.

Fig. R22 Temperature dependent XMCD characterization of 1 ML CrTe₂ film. (a) Typical pairs of XAS and XMCD spectra of 1 ML CrTe₂ thin film at various temperatures, where dichroism at Cr *L*₃ edge can be traced to 200 K (spectra at different

temperatures are offset for clarify). (b) The partial enlarged XAS spectra of 1 ML CrTe₂, where the difference between left- and right-circularly polarized XAS is evident. (c) XMCD percentage versus temperature derived from (a) with the trend guiding lines.

After demonstrating the ferromagnetism in 1 ML CrTe₂ film, we now discuss the electronic structures of CrTe₂ thin films with different thicknesses. To understand the thickness-dependent electronic structure, we carried out first-principles calculations of 1T-CrTe₂ with different thicknesses. As shown in Fig. R23, there is an excellent agreement between our experiment and theory. In particular, the hole-like band near E_F and a relatively flat Cr 3*d* orbital band are similar to that of calculated 1T-CrTe₂ with the inclusion of spin polarization. For the 1 ML film, the two parabolic hole pockets are well reproduced by the majority spin projections of the bands, which highlights the FM nature. These results demonstrate that the epitaxial 1T structure and ferromagnetism have been established since 1 ML deposition.

Fig. R23 Band structures of CrTe₂ ultrathin films. (a) 1 ML. (b) 2 ML. (c) 3 ML. (d) 5 ML. (e) 7 ML. All the spectra were taken along the high symmetry direction M- Γ -M. Upper panels: band structures of 1T phase from first-principles calculations; Middle panels: calculated minority (red) and majority (blue) spin projections of the bands, respectively; Lower panels: ARPES intensity maps.

Revision: Following reviewer’s comments, we have included XMCD study of 1 ML CrTe₂ film in Fig. 4. We have further discussed the relationship between electronic structure and magnetism against thickness variation in the revised manuscript on page 14.

“The magnetic response of 1 ML CrTe₂ film is worth exploring. It is difficult to detect magnetization in such ultrathin films by SQUID, since the magnetic signal of 1 ML CrTe₂ is too weak compared with an overwhelmingly larger background signal from the substrate and beyond the resolution of SQUID. Therefore, we did element-specific XMCD characterization of 1 ML CrTe₂ film (Fig. 4a). There is a clear difference in the XAS spectra between left- and right-handed circularly polarized setups (Fig. 4b). Although the dichroism is small compared with 7 ML sample, the clear XMCD signals appear near the absorption peaks. It suggests that the intrinsic ferromagnetism of 1 ML CrTe₂ film originates from the spin polarization of Cr 3d electrons. Accurate calculation of the magnetic moment remains a challenge since the contribution of Te capping layer to the XAS spectra is so large for 1 ML sample. The XMCD percentage increases with decreasing temperature (Fig. 4c), in line with the typical FM behavior. The nonzero XMCD percentage persists when temperature approaches 200 K and disappears at 250 K, indicating that 1 ML CrTe₂ has a T_C of ~200 K.”

“To understand the thickness-dependent electronic structure, we carried out first-principles calculations of 1T-CrTe₂ with different thicknesses. As shown in Supplementary Fig. 16, there is an excellent agreement between our experiment and theory. In particular, the hole-like band near E_F and a relatively flat Cr 3d orbital band are similar to that of calculated 1T-CrTe₂ with the inclusion of spin polarization. For the 1 ML film, the two parabolic hole pockets are well reproduced by the majority spin projections of the bands, which highlights the FM nature. These results demonstrate that the epitaxial 1T structure and ferromagnetism have been established since 1 ML deposition, in line with the corresponding STM images.”

4, Temperature dependence of magnetization in Fig. 3d is apparently different trend with that in Fig. 2a. Fig. 3d shows monotonic increase with decreasing temperature. However, 7 ML data in Fig. 2a steep increase around 250 K, then keep constant at low temperature. This difference should be discussed. If this difference comes from the measurement configuration, in-plane or out-of-plane, the authors should provide temperature dependence of out-of-plane magnetization. If this comes from thickness difference or measurement method, it should be explained.

Reply: We appreciate the reviewer’s comments. The different trends of temperature dependent magnetization come from the measurement configuration. As shown in Fig. R24, for 7 ML CrTe₂ thin films, the out-of-plane direction is an easy axis. The out-of-plane magnetization shows monotonic increase with decreasing temperature.

Fig. R24 Zero-field and field cooled temperature dependence of the magnetization of 7 ML CrTe₂ with an applied in-plane and out-of-plane magnetic field of 0.1 T.

Moreover, as we put the temperature dependent magnetization curves characterized by SQUID and XMCD together (Fig. R25), both of them show monotonic increase with decreasing temperature.

Fig. R25 The temperature dependent out-of-plane magnetization and m_s of 7 ML CrTe₂ obtained by SQUID and XMCD, respectively.

Revision: According to reviewer’s advices, we have discussed the origin of different temperature dependent behavior of out-of-plane and in-plane magnetization in Fig. S3. “The out-of-plane magnetization increases with decreasing temperature, indicating the enhanced ferromagnetism at low temperature. On the contrary, the slight decrease of in-plane magnetization at low temperature is resulted from the accelerated process of spin reorientation from the ab plane to the c axis, due to the weak in-plane magnetic field.”

5, PMA is compared with Ku and saturation magnetization. Ku should be explained by equation. The magnetization in CrTe₂ is rather small compared to other metal ferromagnets with PMA. Degree of PMA should be discussed on the same measure.

For my sense, “strong PMA” is not appropriate for the observed data of CrTe₂ in this manuscript because the value is not so large. If the authors try to claim the “strong PMA”, please provide the values for famous strong PMA metal ferromagnets to compare.

Reply: The PMA constant (K_u) can be estimated from the relation $K_u = \frac{H_k M_s}{2}$. Apart from the saturated magnetization (M_s), the perpendicular anisotropy field (H_k) is crucial, which can be deduced from the in-plane and out-of-plane magnetization hysteresis loops.

Table R1. Maximal K_u in various strong PMA systems from literatures^a

Material	Maximal K_u (Merg/cm ³)	Ref.
7 ML CrTe ₂ films	5.63	[*]
Fe ₃ GeTe ₂ nanoflakes	1.5	36
CrI ₃ bulk	3	37
Cr ₂ Ge ₂ Te ₆ bulk	0.48	38
Fe ₂ CrSi films	2.8	39
[Co/Pd] ₆ films	3.4	40
[Co/Pt] ₆ films	9	40
L0 ₂₂ -Mn ₃ Ga films	0.89	41
CoFeB films	2.1	14
Co ₂ FeAl films	1.3	42

According to Table R1, the K_u in CrTe₂ film system is comparable to the typical values in famous FM systems, such as Co/Pd and Co/Pt. More remarkably, the K_u is also the highest one among 2D ferromagnets (e.g., Fe₃GeTe₂³⁶, Cr₂Ge₂Te₆³⁸, CrI₃³⁷) reported so far. A strong PMA is of crucial importance to maintain high thermal stability at moderate coercive fields for long-term data retention in high-density magnetic data-storage devices.

Revision: Based on reviewer’s suggestion, we have added the equation of K_u in the revised manuscript on page 7 and the comparison of K_u in CrTe₂ film system with other famous FM systems with strong PMA in supplementary materials (Table S1).

“The sharp distinction between out-of-plane (Fig. 2b) and in-plane (Fig. 2c) M-H loops demonstrates a strong out-of-plane anisotropy of the magnetization with a large PMA constant ($K_u = \frac{H_k M_s}{2}$) of 5.63×10^6 erg/cm³ at 20 K. The K_u in CrTe₂ film system is comparable to the typical values in famous FM systems, such as Co/Pd and Co/Pt, with strong PMA (Supplementary Table 1), which is important for obtaining 2D FM order and is also considerably desirable for vdW heterostructures-based spintronics.”

Comments: 6, please check carefully following minor points.

Comments: 6-1, “doping a FM host with specific elements” in page 3, a word of FM means ferromagnetic. What did the word “a FM host” mean?

Reply: Thanks for the comments. Here, “a FM host” means the host itself is ferromagnetic.

Comments: 6-2, “modulating the magnetic order via electrolyte gating” in page 3, gating can modulate the charge density and/or electric field at the interface, resulting in the variation of magnetism. The word of “magnetic order” has some meanings, which may confuse. Please revise carefully.

Reply: We thank the reviewer for pointing out this. We have now corrected our confusing statement. In the revised manuscript, we have changed this sentence to “*The third method is inducing electron doping via electrolyte gating, and thereby modulating the T_C of ferromagnetism.*”

Comments: 6-3, Refs.10-12 are really appropriate references for the enhancement of FM order by proximity effect?

Reply: We appreciate the reviewer’s comments. In Ref. 10, W. Q. Liu *et al.* reported a study of enhancing the magnetic ordering in $\text{Bi}_{2-x}\text{Cr}_x\text{Se}_3$ ⁴³, via the proximity effect using a high- T_C ferrimagnetic insulator $\text{Y}_3\text{Fe}_5\text{O}_{12}$. An enhanced T_C of 50 K was observed in this magnetically doped TI/FMI heterostructure.

In Ref. 11, the m_1 of Fe at Fe/graphene was enhanced by ~200% compared to m_1 of that in the bulk⁴⁴. The enhancement of m_1 of Fe can be attributed to the symmetry breaking of the ultrathin films, in which the electrons are rather localized around the nucleus, leading to an orbital degeneracy lifting. However, the spin moment is ~50% reduced from the bulk-like Fe, which is attributed to the strong hybridization between the Fe $3d_{z^2}$ and the C $2p_z$ orbitals and the *sp*-orbital-like behavior of the Fe $3d$ electrons due to the presence of graphene. Taken together, Ref. 11 may not be an appropriate reference for the statement of enhanced FM order by proximity effect. Therefore, we have deleted this reference.

We cited a wrong reference for Ref. 12. The paper we want to cite here is “*Nat. Mater.* 16, 94-100 (2017)” by Q. L. He *et al.*⁴⁵. They demonstrated the use of antiferromagnetic exchange coupling in manipulating the magnetic properties of magnetic topological insulators. The AFM-based Proximity effects are shown to induce an interfacial spin texture modulation and establish an effective long-range exchange coupling, which significantly enhances the magnetic ordering temperature in the superlattice.

Revision: We have revised the references in the main text on page 3.

“The second one is constructing heterostructures with FM (or ferrimagnetic) metals (or insulators), in which the FM order can be enhanced by proximity effects^{43,45}.”

Comments: 6-4, “FM order of the CrTe₂ is insensitive to the thickness” in page 6 is difficult to agree, because the judgement whether sensitive or insensitive depends on the system. The plot for T_C of film/T_C of bulk as a function of thickness for CrTe₂ and Cr₂Ge₂Te₆ seem meaningful to compare.

Reply: We thank the reviewer for this suggestion. The normalized T_C as a function of thickness is shown in Fig. R26. Compared with Fe₃GeTe₂⁷ and Cr₂Ge₂Te₆², the T_C of CrTe₂ displays a weaker dependence on the number of layers.

Fig. R26 T_C (normalized to bulk T_C for the particular material) as a function of the number of layers. Data for Fe₃GeTe₂ adapted from ref. ⁷; for Cr₂Ge₂Te₆, ref. ².

Revision: According to reviewer’s suggestions, we have changed this sentence into “*In order to investigate the dimensionality effect of the ferromagnetism in CrTe₂ stemming from thermal fluctuation, we plot the thickness dependent T_C (Fig. 4d). The T_C of CrTe₂ decreases slightly with reducing the film thickness compared with other commonly investigated 2D magnets, such as Cr₂Ge₂Te₆² and Fe₃GeTe₂⁷ (Supplementary Fig. 8), demonstrating the robustness of ferromagnetism in the epitaxially grown CrTe₂ thin films.*”

Comments: 6-5, What does “quantum thickness regime” in page 7 mean? How did the authors define and evaluate the quantum feature?

Reply: Here, we want to express “few atomic layers”. We agree that using “quantum thickness regime” may be not appropriate, and have replaced it with “atomic layers”.

Comments: 6-6, “large magnetic moments” in page 12 is not discussed with the exact values for the films in main text. It is difficult to judge the large or small moment.

Revision: Based on reviewer’s suggestion, we have added the discussion of large magnetic moment in the revised manuscript on page 10.

“Notably, the atomic magnetic moment of CrTe₂ is determined to be ~3 μ_B/atom with the half-filled t_{2g} orbital. It is the largest possible moment of Cr according to the Hund’s rule.”

Comments: 6-7, “the thickness down to 3 ML due to the strong magnetic anisotropy” in page 12 is not supported by experimental results, because the M-H curves of in-plane and out-of-plane for 3 and 5 ML films are not provided in the present manuscript. PMA should be judged by comparison of M-H curves of in-plane and out-of-plane.

Reply: We thank the reviewer for suggesting the additional experiments. In order to clarify the PMA in ultrathin CrTe₂ thin films, we did field dependent magnetization curves of 3 ML and 5 ML CrTe₂ thin films under out-of-plane and in-plane configuration (Fig. R27). The nearly square-shaped FM hysteresis loops under out-of-plane magnetic field suggest the robust FM order with the easy axis perpendicular to the thin films. The PMA constants ($K_u = \frac{H_k M_s}{2}$) of 3 ML and 5 ML CrTe₂ films at 10 K are determined to be 6.6×10⁶ erg/cm³ and 6.5×10⁶ erg/cm³, respectively.

Fig. R27 M-H loops of CrTe₂ thin films with different thicknesses. a-d) Temperature dependent magnetic hysteresis loops of 3 ML (a,c) and 5 ML (b,d) CrTe₂ thin films with out-of-plane and in-plane field, indicating a strong out-of-plane magnetic anisotropy.

Revision: Following reviewer’s comments, we have included the magnetic hysteresis

loops of 3 ML and 5 ML CrTe₂ thin films as supplementary materials (Fig. S5).

“In order to clarify the thickness dependence of the magnetic properties, we have measured the field dependent magnetization curves of 3 ML and 5 ML CrTe₂ thin films under out-of-plane and in-plane configuration (Supplementary Fig. 5). The nearly square-shaped FM hysteresis loops under out-of-plane magnetic field suggest the robust FM order with the easy axis perpendicular to the thin films. At 10 K, the magnetic moments of 3 ML and 5 ML CrTe₂ are found to be 2.81 μ_B /Cr and 2.83 μ_B /Cr, respectively, which are comparable with the theoretical value ($\sim 3 \mu_B$ /Cr)²⁴. With in-plane and out-of-plane M-H loops, the PMA constants of 3 ML and 5 ML CrTe₂ films are determined to be 6.6×10^6 erg/cm³ and 6.5×10^6 erg/cm³ at 10 K, respectively.”

Comments: 6-8, Growth rate and provided flux rate should be addressed for general readers.

Revision: Based on the reviewer’s suggestion, we have addressed the growth rate and flux rate in Methods on page 15.

“Then, high-purity Cr and Te were evaporated from an electron-beam evaporator and a standard Knudsen cell, with flux of 0.1 \AA /min and 6 \AA /min, respectively.” “The deposition rate of CrTe₂ was ~ 0.73 \AA /min as monitored by a quartz oscillator.”

Comments: 6-9, There is a paper not referred in this manuscript, Room temperature ferromagnetism in ultra-thin van der Waals crystals of 1T-CrTe₂, Nano Research. Doi.org/10.1007/s12274-020-3021-4.

Reply: We thank the reviewer for pointing out this interesting paper. When we submit this manuscript, the paper (Nano Research. Doi.org/10.1007/s12274-020-3021-4)¹⁶ hasn’t been published online and we only refer it in the format of *arXiv* as ref.22 in our first submitted manuscript.

Revision: The paper of Nano Research¹⁶ has now been referenced as a published one (Nano Research. Doi.org/10.1007/s12274-020-3021-4) in our revised manuscript on page 5 and 8.

In addition to the above responses to all three reviewers’ comments, we also have made the following revisions.

1. The figures and references have been updated.
2. Author Prof. Jun Du is included in the author list. He provided us with further SQUID measurements and valuable suggestions.

References

- 1 Huang, B. *et al.* Layer-dependent ferromagnetism in a van der Waals crystal down to the monolayer limit. *Nature* **546**, 270-273 (2017).
- 2 Gong, C. *et al.* Discovery of intrinsic ferromagnetism in two-dimensional van der Waals crystals. *Nature* **546**, 265-269 (2017).
- 3 McGuire, M. A. *et al.* Magnetic behavior and spin-lattice coupling in cleavable van der Waals layered CrCl₃ crystals. *Phys. Rev. Mater.* **1**, 014001 (2017).
- 4 Casto, L. D. *et al.* Strong spin-lattice coupling in CrSiTe₃. *APL Mater.* **3**, 041515 (2015).
- 5 Chen, S. *et al.* Boosting the Curie Temperature of Two-Dimensional Semiconducting CrI₃ Monolayer through van der Waals Heterostructures. *J. Phys. Chem. C* **123**, 17987-17993 (2019).
- 6 Bastos, C. M. O., Besse, R., Da Silva, J. L. F. & Sipahi, G. M. Ab initio investigation of structural stability and exfoliation energies in transition metal dichalcogenides based on Ti-, V-, and Mo-group elements. *Phys. Rev. Mater.* **3**, 044002 (2019).
- 7 Deng, Y. *et al.* Gate-tunable room-temperature ferromagnetism in two-dimensional Fe₃GeTe₂. *Nature* **563**, 94-99 (2018).
- 8 Freitas, C. D. *et al.* Ferromagnetism in layered metastable 1T-CrTe₂. *J. Phys.: Condens. Matter* **27**, 176002 (2015).
- 9 Purbawati, A. *et al.* In-plane magnetic domains and Neel-like domain walls in thin flakes of the room temperature CrTe₂ van der Waals ferromagnet. *ACS Appl. Mater. Inter.* **12**, 30702-30710 (2020).
- 10 Lau, Y.-C. *et al.* Giant perpendicular magnetic anisotropy in Ir/Co/Pt multilayers. *Phys. Rev. Mater.* **3**, 104419 (2019).
- 11 Lado, J. L. & Fernández-Rossier, J. On the origin of magnetic anisotropy in two dimensional CrI₃. *2D Mater.* **4**, 035002 (2017).
- 12 Fujisawa, Y. *et al.* Tailoring magnetism in self-intercalated Cr_{1+δ}Te₂ epitaxial films. *Phys. Rev. Mater.* **4**, 114001 (2020).
- 13 Lee, A. J. *et al.* Interfacial Rashba-Effect-Induced Anisotropy in Nonmagnetic-Material-Ferrimagnetic-Insulator Bilayers. *Phys. Rev. Lett.* **124**, 257202 (2020).
- 14 Ikeda, S. *et al.* A perpendicular-anisotropy CoFeB-MgO magnetic tunnel junction. *Nat. Mater.* **9**, 721-724 (2010).
- 15 Dieny, B. & Chshiev, M. Perpendicular magnetic anisotropy at transition metal/oxide interfaces and applications. *Rev. Mod. Phys.* **89**, 025008 (2017).
- 16 Sun, X. *et al.* Room temperature ferromagnetism in ultra-thin van der Waals crystals of 1T-CrTe₂. *Nano Res.* **13**, 3358-3363 (2020).
- 17 Ye, M. *et al.* Carrier-mediated ferromagnetism in the magnetic topological insulator Cr-doped (Sb,Bi)₂Te₃. *Nat. Commun.* **6**, 8913 (2015).
- 18 Wong, P. K. J. *et al.* Evidence of Spin Frustration in a Vanadium Diselenide Monolayer Magnet. *Adv. Mater.* **31**, 1901185 (2019).
- 19 Tang, S. *et al.* Quantum spin Hall state in monolayer 1T'-WTe₂. *Nat. Phys.* **13**, 683-687 (2017).
- 20 Yun Zhang *et al.* Emergence of Kondo lattice behavior in a van der Waals itinerant ferromagnet, Fe₃GeTe₂. *Sci. Adv.* **4**, eaao6791 (2018).
- 21 Li, Y. F. *et al.* Electronic structure of ferromagnetic semiconductor CrGeTe₃ by angle-resolved photoemission spectroscopy. *Phys. Rev. B* **98**, 125127 (2018).

- 22 Lazar, S. *et al.* Imaging, core-loss, and low-loss electron-energy-loss spectroscopy mapping in
aberration-corrected STEM. *Microsc. Microanal.* **16**, 416-424 (2010).
- 23 Chen, J. *et al.* Evidence for Magnetic Skyrmions at the Interface of Ferromagnet/Topological-
Insulator Heterostructures. *Nano Lett.* **19**, 6144-6151 (2019).
- 24 Otero Fumega, A., Phillips, J. & Pardo, V. Controlled Two-Dimensional Ferromagnetism in 1T-
CrTe₂: The Role of Charge Density Wave and Strain. *J. Phys. Chem. C* **124**, 21047-21053 (2020).
- 25 Noh, H. J. *et al.* Valence values of the cations in selenospinel Cu(Cr,Ti)₂Se₄. *EPL* **78**, 27004
(2007).
- 26 Zhao, W. *et al.* Metastable MoS₂: Crystal Structure, Electronic Band Structure, Synthetic
Approach and Intriguing Physical Properties. *Chemistry* **24**, 15942-15954 (2018).
- 27 Lv, Y. Y. *et al.* Composition and temperature-dependent phase transition in miscible Mo_{1-x}W_xTe₂
single crystals. *Sci. Rep.* **7**, 44587 (2017).
- 28 Hu, T., Li, R. & Dong, J. Characterization of few-layer 1T-MoSe₂ and its superior performance
in the visible-light induced hydrogen evolution reaction. *J. Chem. Phys.* **139**, 174702 (2013).
- 29 Wang, C. *et al.* Bethe-Slater-curve-like behavior and interlayer spin-exchange coupling
mechanisms in two-dimensional magnetic bilayers. *Phys. Rev. B* **102**, 020402 (2020).
- 30 Wang, Q., Sun, S., Zhang, X., Pang, F. & Lei, H. Anomalous Hall effect in a ferromagnetic
Fe₃Sn₂ single crystal with a geometrically frustrated Fe bilayer kagome lattice. *Phys. Rev. B* **94**,
075135 (2016).
- 31 Liu, W. *et al.* Experimental Observation of Dual Magnetic States in Topological Insulators. *Sci.*
Adv. **5**, eaav2088 (2019).
- 32 Shabbir, B. *et al.* Long range intrinsic ferromagnetism in two dimensional materials and
dissipationless future technologies. *Appl. Phys. Rev.* **5**, 041105 (2018).
- 33 Zhang, J. *et al.* Unveiling Electronic Correlation and the Ferromagnetic Superexchange
Mechanism in the van der Waals Crystal CrSiTe₃. *Phys. Rev. Lett.* **123**, 047203 (2019).
- 34 May, A. F., Calder, S., Cantoni, C., Cao, H. & McGuire, M. A. Magnetic structure and phase
stability of the van der Waals bonded ferromagnet Fe_{3-x}GeTe₂. *Phys. Rev. B* **93**, 014411 (2016).
- 35 Harrison, R. G. Calculating the spontaneous magnetization and defining the Curie temperature
using a positive-feedback model. *J. Appl. Phys.* **115**, 033901 (2014).
- 36 Kim, D. *et al.* Antiferromagnetic coupling of van der Waals ferromagnetic Fe₃GeTe₂.
Nanotechnology **30**, 245701 (2019).
- 37 Richter, N. *et al.* Temperature-dependent magnetic anisotropy in the layered magnetic
semiconductors CrI₃ and CrBr₃. *Phys. Rev. Mater.* **2**, 024004 (2018).
- 38 Zeisner, J. *et al.* Magnetic anisotropy and spin-polarized two-dimensional electron gas in the
van der Waals ferromagnet Cr₂Ge₂Te₆. *Phys. Rev. B* **99**, 165109 (2019).
- 39 Wang, Y.-P. *et al.* Perpendicular magnetic anisotropy in Fe₂Cr_{1-x}Co_xSi Heusler alloy. *J. Phys.*
D: Appl. Phys. **47**, 495002 (2014).
- 40 Yakushiji, K. *et al.* Ultrathin Co/Pt and Co/Pd superlattice films for MgO-based perpendicular
magnetic tunnel junctions. *Appl. Phys. Lett.* **97**, 232508 (2010).
- 41 Kurt, H., Rode, K., Venkatesan, M., Stamenov, P. & Coey, J. M. D. High spin polarization in
epitaxial films of ferrimagnetic Mn₃Ga. *Phys. Rev. B* **83**, 020405 (2011).
- 42 Wu, Y., Xu, X. G., Miao, J. & Jiang, Y. Perpendicular Magnetic Anisotropy in Co-Based Full
Heusler Alloy Thin Films. *Spin* **5**, 1540012 (2016).
- 43 Liu, W. *et al.* Enhancing magnetic ordering in Cr-doped Bi₂Se₃ using high-T_C ferrimagnetic

- insulator. *Nano Lett.* **15**, 764-769 (2015).
- 44 Liu, W. Q. *et al.* Atomic-Scale Interfacial Magnetism in Fe/Graphene Heterojunction. *Sci. Rep.* **5**, 11911 (2015).
- 45 He, Q. L. *et al.* Tailoring exchange couplings in magnetic topological-insulator/antiferromagnet heterostructures. *Nat. Mater.* **16**, 94-100 (2017).

Reviewers' Comments:

Reviewer #1:

Remarks to the Author:

The Authors have carefully taken my suggestions and comments into account. They have provided convincing answers and adequately modified their manuscript. I have nothing to oppose to the publication of the paper.

Reviewer #2:

Remarks to the Author:

It seems that authors answered all the questions appropriately and improved the manuscript by adding new discussion. I would like to confirm one additional point about the perpendicular magnetic anisotropy (related to the question 7, 8, 9, and 10 in the previous report). In Fig. 2 c and d, we can see hysteresis even for H||ab data, which looks inconsistent with the perpendicular magnetic anisotropy. Could you explain the possible origin of this hysteresis in case of H||ab?

Reviewer #3:

Remarks to the Author:

Dear Authors,

I appreciate the Authors for the revisions and responses. I agree with the significance of this study, which is clearly raised in the response. Two experimental observations are pronounced in the revised manuscript; 1) discussion about the PMA and 2) observation of ferromagnetic behavior in monolayer CrTe₂.

The responses to my comments and revisions are basically reasonable.

Please check again following remained points, which should be clarified.

Relating on previous comment 1, title

Thanks for the revisions regarding the definition about intrinsic ferromagnetism.

In the revised title, what does [atomic-layer] mean? It is rather difficult to find the definition of [atomic-layer films] as adjective. I strongly recommend to remove the [atomic-layer] from title.

Relating on previous comment 2, Regarding with Fig. R17.

Does the magnetization reach zero at high temperature for the data shown in Fig. R17?

How did the Authors define Curie temperature for the data H//ab and H//c in Fig. R17?

As I pointed out the two-step like transition of M-T curves in previous comment, the data in Fig. R17 also shows two-step and saturation at finite value around 300 K. This behavior is completely different from the data in Fig. R16, which reaches zero. In addition, are Curie temperatures estimated from the data H//ab and H//c consistently comparable in the identical sample?

The Fig. R17 corresponds to Supplementary Fig. 3 so that the above points should be clarified. I recommend to add the discussion and response to the above comment in Supplementary file.

In Fig. 2a, the measurement configuration of film, direction of magnetic field, and the value of magnetic field are confusing. Therefore, the schematic of the measurement configuration should be added in Fig. 2a as like Fig. 2b and Fig. 2c for removing the confusion.

Relating on previous comment 3, Fig. 2b, Fig. R20, and Fig. R21.

It is difficult to agree with the claim in Fig. R20. The authors have to provide fair comparison with coercive field in M-H and Ryx-H curves at identical temperature. For example, Rxy-H and M-H curve at 10 K or 100 K should be directly compared within one figure, which enables understanding the comparable coercive field easily. This figure can be individually made for 3ML, 5ML, and 7ML.

In Fig. R20a, 20b, and 20c, Ryx-H curves at $T = 10$ K is missing, although the data point is plotted in Fig. R20d, 20e, and 20f. Considering from Ryx-H curves in Figs. R20 at high temperature $T = 30$ K and 70 K, the coercive field in Ryx- H curves at $T = 10$ K is likely much larger than that in M-H curves in Fig. 2b and Fig. R21. In addition, the number of data points in Figs. R20d, 20e, and 20f does not match the data shown in Figs. R20a, 20b, and 20c. For example, 70 K data for AHE shown in Figs. R20a, 20b, and 20c is not included in Figs. R20d, R20e, and R20f. Please revise the figures consistently.

Moreover, logarithmic plot of Figs. R20d, 20e, and 20f has to be revised to linear scale. The important comparison should be focused on low temperature region in particular lower than 100 K. Linear scale is preferable to compare it. Probably the authors know well that the shape of Ryx-H curves are recognized as identical to the M-H curves. The authors have to provide fair comparison. If the coercive field of Ryx-H and M-H is not consistent each other, the authors have to explain the reason.

These figures of Figs. R20 and Fig. R21 correspond to Supplementary Figures 5 and 6 so that the above point should be clarified. I recommend to add the discussion and response to the above comment in Supplementary file.

In the revised description in page 6 line 18.

[Both TEM and STM characterizations manifest our as-grown films are stoichiometric CrTe_2 .] is added. However, strictly speaking, it is quite difficult to obtain [stoichiometric] film by MBE. I agree that the chemical composition of the film is close to $\text{Cr} : \text{Te} = 1 : 2$, but it is rather difficult to conclude perfect stoichiometry as like $\text{Cr} : \text{Te} = 1.000 : 2.000$. The expression of [Stoichiometric film] corresponds to no point defect with totally-matched composition of Cr and Te. I recommend to revise more acceptable description.

In the revised description in page 6, line 20.

[$c = 6.13$ for the film and $c = 5.94$ for the bulk] is added. Roughly 3 % mismatch of c-axis length is rather large. In-plane lattice of the film agrees well with the bulk value, which is discussed by STM image in page 6, line 4. This in-plane value indicates small effect of in-plane strain in the film. The Authors have to discuss the origin why the c-axis length is so large. If the a-axis length is elongated 3 %, band structure is dramatically varied from bulk. In addition, is this elongation of c-axis length relating on the PMA? Because the magnetic exchange coupling between two layers strongly depends on the length, the magnetic property of CrTe_2 , CrSe_2 , and CrS_2 are completely different. I recommend to the authors do not overlook the large elongation of c-axis length.

Reply to the referees' comments

We thank the referees for the helpful and positive comments and have revised the paper accordingly to address the points raised. We show reviewers' reports in black typeface and answers in light blue typeface for easy identification. Major changes have been highlighted in blue in the revised main text and the supplementary information.

Reviewer #1 (Remarks to the Author):

Comments:

The Authors have carefully taken my suggestions and comments into account. They have provided convincing answers and adequately modified their manuscript. I have nothing to oppose to the publication of the paper.

Reply: We thank the reviewer for his/her positive appraisal and publication recommendation.

Reviewer #2 (Remarks to the Author):

Comments:

It seems that authors answered all the questions appropriately and improved the manuscript by adding new discussion. I would like to confirm one additional point about the perpendicular magnetic anisotropy (related to the question 7, 8, 9, and 10 in the previous report). In Fig. 2 c and d, we can see hysteresis even for H//ab data, which looks inconsistent with the perpendicular magnetic anisotropy. Could you explain the possible origin of this hysteresis in case of H//ab?

Reply: We thank the reviewer for his/her positive comments and he/she thinks that we have answered all the questions appropriately and improved the manuscript. The magnetic anisotropy generally originates from a combination of three factors: shape anisotropy^{1,2}, magneto-crystalline anisotropy³ and anisotropy of exchange interactions¹. The shape anisotropy favors in-plane magnetic anisotropy in thin films^{1,2}. The magneto-crystalline anisotropy and exchange anisotropy lead to the PMA in CrTe₂ films. However, the in-plane shape anisotropy is also present in the thin films, which gives rise to the in-plane hysteresis. Similar in-plane magnetic hysteresis loops have also been reported in the ferromagnetic vdW Cr₂Ge₂Te₆ thin films⁴ and typical PMA systems such as Mn_{2.5}Ga⁵ and Co/Pt⁶.

Revision: According to the reviewer's suggestions, we have included a brief discussion of the in-plane magnetic hysteresis loops in the revised manuscript on page 7.

"The in-plane magnetic hysteresis loops, similar to those reported in the FM vdW

Cr₂Ge₂Te₆ thin films⁴ and typical PMA systems such as Mn_{2.5}Ga⁵ and Co/Pt⁶, can be attributed to the shape anisotropy favoring in-plane easy axis for thin films^{1,2}.”

Reviewer #3 (Remarks to the Author):

Comments:

Dear Authors,

I appreciate the Authors for the revisions and responses. I agree with the significance of this study, which is clearly raised in the response. Two experimental observations are pronounced in the revised manuscript; 1) discussion about the PMA and 2) observation of ferromagnetic behavior in monolayer CrTe₂.

The responses to my comments and revisions are basically reasonable.

Reply: First of all, we are grateful to the reviewer for pointing out the significance of this study and the revisions we made. We also thank the reviewer for the detailed comments and constructive suggestions, which help us to further improve our manuscript.

Comments: Please check again following remained points, which should be clarified.

Relating on previous comment 1, title

Thanks for the revisions regarding the definition about intrinsic ferromagnetism.

In the revised title, what does [atomic-layer] mean? It is rather difficult to find the definition of [atomic-layer films] as adjective. I strongly recommend to remove the [atomic-layer] from title.

Reply: We thank the reviewer for this valuable comment. In the revised manuscript, the title has been modified to: “*Room-temperature intrinsic ferromagnetism in epitaxial CrTe₂ ultrathin films.*”

Comments: Relating on previous comment 2, Regarding with Fig. R17.

Does the magnetization reach zero at high temperature for the data shown in Fig. R17? How did the Authors define Curie temperature for the data H//ab and H//c in Fig. R17? As I pointed out the two-step like transition of M-T curves in previous comment, the data in Fig. R17 also shows two-step and saturation at finite value around 300 K. This behavior is completely different from the data in Fig. R16, which reaches zero. In addition, are Curie temperatures estimated from the data H//ab and H//c consistently comparable in the identical sample?

The Fig. R17 corresponds to Supplementary Fig. 3 so that the above points should be clarified. I recommend to add the discussion and response to the above comment in Supplementary file.

Reply: We appreciate the reviewer's comments and suggestions. Previously we plotted the magnetization curves only up to 300K in Fig.R17. Here we have revised the figure to show the out-of-plane and in-plane magnetization curves up to 320 K, as exhibited in Fig. R1. Both magnetization curves show a smaller slope near T_C and reach zero at ~ 320 K.

Fig. R1 Zero-field and field cooled temperature dependence of the magnetization of 7 ML CrTe₂ with an applied in-plane and out-of-plane magnetic field of 0.1 T.

The T_C has been obtained by using a critical power-law function $\alpha(1-T/T_C)^\beta$ to fit M-T curves without the inclusion of the paramagnetic tail. This method had been used to study ferromagnetism in atomically-thin Cr₂Ge₂Te₆⁷ and Fe₃GeTe₂ nanoflakes⁸. As shown in Fig. R2, the T_C of 7 ML CrTe₂ thin films extracted from the temperature-dependent out-of-plane and in-plane magnetization are $\sim 300 \pm 5$ K and 268 ± 3 K with β of 0.13 ± 0.03 and 0.11 ± 0.04 , respectively, which is comparable with $\beta = 0.125$ for the 2D Ising model⁸. The T_C calculated along the hard axis is lower than that of the easy axis. The theoretical work by Callen⁹ shows that for an anisotropic ferromagnet, the T_C depends on the direction of the magnetization. The T_C is high along the easy directions, and can drop quite low in hard directions, for an anisotropy energy comparable to the exchange energy. Distinct with 3D materials in which the typical value of exchange interaction is orders of magnitudes larger than magnetic anisotropy, the T_C in 2D ferromagnets is determined primarily by the excitation gap that results from the magnetic anisotropy⁷. Therefore, the dominant magnetic anisotropy in 2D magnetic materials gives rise to the variation of T_C along different magnetic directions. Similar behaviors were also observed in Fe₃GeTe₂ (201 K along the easy axis and 196 K along the hard axis)¹⁰ and Fe₇S₈ single crystals (603 K along the easy axis and 225 K along the hard axis)¹¹.

Fig. R2 Criticality analysis for 7 ML CrTe₂ thin films with out-of-plane (0.1 T) and in-plane magnetic field (0.5 T). The solid lines are the fitting curves with the form of $\alpha(1-T/T_C)^\beta$.

Revision: Per the reviewer’s advice, we have replaced the $M-T$ curve in Fig. S3 with the new one with temperature up to 320 K and discussed the definition of the T_C in the manuscript as follows.

“The T_C has been obtained by using a critical power-law function $\alpha(1-T/T_C)^\beta$ to fit $M-T$ curves without the inclusion of the paramagnetic tail⁸.

In the supplementary, we have added the following paragraph.

“The T_C of 7 ML CrTe₂ thin films extracted from the temperature-dependent out-of-plane and in-plane magnetization are $\sim 300 \pm 5$ K and 268 ± 3 K with β of 0.13 ± 0.03 and 0.11 ± 0.04 , respectively, which is comparable with $\beta = 0.125$ for the 2D Ising model⁸. The T_C calculated along the hard axis is lower than that of the easy axis. The theoretical work by Callen⁹ shows that for an anisotropic ferromagnet, the T_C depends on the direction of the magnetization for an anisotropy energy comparable to the exchange energy. Distinct with 3D materials in which the typical value of exchange interaction is orders of magnitudes larger than magnetic anisotropy, the T_C in 2D ferromagnets is determined primarily by the excitation gap that results from the magnetic anisotropy⁷. Therefore, the dominant magnetic anisotropy in 2D magnetic materials gives rise to the variation of T_C along different magnetic directions. Similar behaviors were also observed in Fe₃GeTe₂ (201 K along the easy axis and 196 K along the hard axis)¹⁰ and Fe₇S₈ single crystals (603 K along the easy axis and 225 K along the hard axis)¹¹.”

Comments: In Fig.2a, the measurement configuration of film, direction of magnetic field, and the value of magnetic field are confusing. Therefore, the schematic of the measurement configuration should be added in Fig. 2a as like Fig. 2b and Fig. 2c for removing the confusion.

Revision: Per the reviewer’s suggestion, we have added the schematic of the measurement configuration to Fig. 2a and the description of the direction and

magnitude of magnetic field to the figure caption.

“The magnetic field is applied along out-of-plane direction with a magnitude of 1000 Oe.”

The new figure is shown below,

Fig. R3 SQUID measurements of the CrTe₂ films. **a** Temperature dependent magnetization curves of the films with various thicknesses under field-cooled mode. The magnetic field is applied along the out-of-plane direction with a magnitude of 1000 Oe. The high T_C is preserved with thickness decreasing to 3 ML. **b**, **c** Magnetic hysteresis loops of 7 ML CrTe₂ at different temperatures with external fields along both the perpendicular (**b**) and parallel orientation (**c**) with respect to sample plane, indicating a strong out-of-plane magnetic anisotropy. **d** Enlarged hysteresis loops of 7 ML CrTe₂ at 300 K, in which the intrinsic ferromagnetism and PMA still maintains. Top inset: temperature dependence of K_u for 7 ML CrTe₂, where the K_u is preserved at 300 K, despite the lower intensity with the increase of temperature.

Comments: Relating on previous comment 3, Fig. 2b, Fig. R20, and Fig. R21.

It is difficult to agree with the claim in Fig. R20. The authors have to provide fair comparison with coercive field in M-H and R_{yx} -H curves at identical temperature. For example, R_{xy} -H and M-H curve at 10 K or 100 K should be directly compared within one figure, which enables understanding the comparable coercive field easily. This figure can be individually made for 3ML, 5ML, and 7ML.

In Fig. R20a, 20b, and 20c, R_{yx} -H curves at $T = 10$ K is missing, although the data point

is plotted in Fig. R20d, 20e, and 20f. Considering from R_{yx} -H curves in Figs. R20 at high temperature $T = 30$ K and 70 K, the coercive field in R_{yx} -H curves at $T = 10$ K is likely much larger than that in M-H curves in Fig. 2b and Fig. R21. In addition, the number of data points in Figs. R20d, 20e, and 20f does not match the data shown in Figs. R20a, 20b, and 20c. For example, 70 K data for AHE shown in Figs. R20a, 20b, and 20c is not included in Figs. R20d, R20e, and R20f. Please revise the figures consistently.

Moreover, logarithmic plot of Figs. R20d, 20e, and 20f has to be revised to linear scale. The important comparison should be focused on low temperature region in particular lower than 100 K. Linear scale is preferable to compare it. Probably the authors know well that the shape of R_{yx} -H curves are recognized as identical to the M-H curves. The authors have to provide fair comparison. If the coercive field of R_{yx} -H and M-H is not consistent each other, the authors have to explain the reason.

These figures of Figs. R20 and Fig. R21 correspond to Supplementary Figures 5 and 6 so that the above point should be clarified. I recommend to add the discussion and response to the above comment in Supplementary file.

Reply: We thank the reviewer for the helpful comments. Per the reviewer's suggestions, we have provided a complete set of Hall resistance plots for a comparison with the coercive field in M-H curves at each temperature. The field dependent magnetic moment and Hall resistance of CrTe₂ thin films of 3 ML, 5 ML and 7 ML at various temperatures are plotted in Fig. R4. The Hall responses share a similar temperature dependence of the coercive field with magnetic hysteresis loops, see Figure R5.

Fig. R4 Comparison between R_{xy} -H and M-H curves of CrTe₂ thin films with thickness of (a-d) 3 ML, (e-f) 5 ML and (i-l) 7 ML at each temperature.

Fig. R5 Comparison of coercivities measured by AHE and M - H loops, which are consistent with each other.

Revision: Following the reviewer’s comments, we have included the magnetic hysteresis loops and electrical transport measurements of CrTe_2 thin films in supplementary materials (Fig. S5, S6 and S7), and added the following discussion.

“The field dependent Hall resistance of CrTe_2 thin films with thickness of (a-d) 3 ML, (e-f) 5 ML and (i-l) 7 ML at various temperatures share a similar trend with the magnetic hysteresis loops.”

“The coercive fields extracted from the Hall responses and M - H loops are consistent with each other.”

Comments: In the revised description in page 6 line 18.

[Both TEM and STM characterizations manifest our as-grown films are stoichiometric CrTe_2 .] is added. However, strictly speaking, it is quite difficult to obtain [stoichiometric] film by MBE. I agree that the chemical composition of the film is close to $\text{Cr} : \text{Te} = 1 : 2$, but it is rather difficult to conclude perfect stoichiometry as like $\text{Cr} : \text{Te} = 1.000 : 2.000$. The expression of [Stoichiometric film] corresponds to no point defect with totally-matched composition of Cr and Te. I recommend to revise more acceptable description.

Revision: We thank the reviewer for this nice advice. We agree with the reviewer that it is technically challenging to obtain stoichiometric films by MBE. We have corrected this statement in the revised manuscript on page 6.

“Both TEM and STM characterizations manifest the epitaxial nature and crystallographic orientation of as-grown CrTe_2 films.”

Comments: In the revised description in page 6, line 20.

[$c = 6.13$ for the film and $c = 5.94$ for the bulk] is added. Roughly 3% mismatch of c -axis length is rather large. In-plane lattice of the film agrees well with the bulk value, which is discussed by STM image in page 6, line 4. This in-plane value indicates small effect of in-plane strain in the film. The Authors have to discuss the origin why the c -axis length is so large. If the a -axis length is elongated 3%, band structure is dramatically varied from bulk. In addition, is this elongation of c -axis length relating on the PMA? Because the magnetic exchange coupling between two layers strongly depends on the length, the magnetic property of CrTe_2 , CrSe_2 , and CrS_2 are completely different. I recommend to the authors do not overlook the large elongation of c -axis

length.

Reply: We thank the reviewer for the valuable suggestions. Actually, $c = 5.94 \text{ \AA}$ for the bulk 1T structure is a theoretically calculated value¹². The experimental lattice parameters of bulk CrTe₂ reported by Freitas *et al*¹³ are $a = 3.79 \text{ \AA}$ and $c = 6.10 \text{ \AA}$, which are close to (within a 0.5% mismatch) the lattice constants of thin films ($c = 6.13 \text{ \AA}$) measured in this work. The reason of inconsistency between theoretical calculation ($c = 5.94 \text{ \AA}$) and experimental results ($c = 6.10 \sim 6.13 \text{ \AA}$) may be due to the fact that spin-orbit coupling (SOC) effects were neglected during the calculation of the structural properties.

We agree with the reviewer that the magnetic exchange coupling is sensitive to the lattice constants. According to the experimental results reported by Freitas *et al*¹⁴, bulk 1T-CrSe₂ with $a = 3.39 \text{ \AA}$ and $c = 5.92 \text{ \AA}$ shows an antiferromagnetic order.

Revision: Per the reviewer's suggestions, we have modified the statement and discussed the dependence of the magnetic exchange coupling on the lattice constants.
"The diffraction pattern with perpendicular constant $c = 6.13 \text{ \AA}$ is matched to the (001) crystal planes of 1T-type hexagonal structure explored experimentally ($a = 3.79 \text{ \AA}$, $c = 6.10 \text{ \AA}$)¹³, rather than those of the 2H phase ($a = 3.49 \text{ \AA}$, $c = 13.64 \text{ \AA}$)¹²."
"We note that the magnetic exchange coupling is sensitive to the lattice parameters. For example, bulk 1T-CrSe₂ with lattice constants of $a = 3.39 \text{ \AA}$ and $c = 5.92 \text{ \AA}$ shows an antiferromagnetic (AFM) order¹⁴, in contrast to the FM phase in CrTe₂."

In addition to the above responses to all three reviewers' comments, we also have made the following revisions.

1. We have carefully reviewed again the manuscript to further improve its English style.
2. The figures and references have been updated.
3. Author Dr. Xiaoqing He is included in the author list. He performed TEM measurements on the samples.

References

- 1 Lado, J. L. & Fernández-Rossier, J. On the origin of magnetic anisotropy in two dimensional CrI₃. *2D Mater.* **4**, 035002 (2017).
- 2 Lau, Y.-C. *et al.* Giant perpendicular magnetic anisotropy in Ir/Co/Pt multilayers. *Phys. Rev. Mater.* **3**, 104419 (2019).
- 3 Fujisawa, Y. *et al.* Tailoring magnetism in self-intercalated Cr_{1+δ}Te₂ epitaxial films. *Phys. Rev. Mater.* **4**, 114001 (2020).
- 4 Mogi, M. *et al.* Ferromagnetic insulator Cr₂Ge₂Te₆ thin films with perpendicular remanence. *APL Mater.* **6**, 091104 (2018).
- 5 Wu, F. *et al.* Epitaxial Mn_{2.5}Ga thin films with giant perpendicular magnetic anisotropy for spintronic devices. *Appl. Phys. Lett.* **94**, 122503 (2009).
- 6 Emori, S. & Beach, G. S. D. Optimization of out-of-plane magnetized Co/Pt multilayers with resistive buffer layers. *J. Appl. Phys.* **110**, 033919 (2011).
- 7 Gong, C. *et al.* Discovery of intrinsic ferromagnetism in two-dimensional van der Waals crystals. *Nature* **546**, 265-269 (2017).
- 8 Fei, Z. *et al.* Two-dimensional itinerant ferromagnetism in atomically thin Fe₃GeTe₂. *Nat. Mater.* **17**, 778-782 (2018).
- 9 Callen, E. R. Anisotropic Curie Temperature. *Phys. Rev.* **124**, 1373-1379 (1961).
- 10 Wang, Y. *et al.* Anisotropic anomalous Hall effect in triangular itinerant ferromagnet Fe₃GeTe₂. *Phys. Rev. B* **96**, 134428 (2017).
- 11 Armstrong, J. N., Hua, S. Z. & Chopra, H. D. Anisotropic Curie temperature materials. *Phys. Status Solidi B* **250**, 387-395 (2013).
- 12 Bastos, C. M. O., Besse, R., Da Silva, J. L. F. & Sipahi, G. M. Ab initio investigation of structural stability and exfoliation energies in transition metal dichalcogenides based on Ti-, V-, and Mo-group elements. *Phys. Rev. Mater.* **3**, 044002 (2019).
- 13 Freitas, C. D. *et al.* Ferromagnetism in layered metastable 1T-CrTe₂. *J. Phys.: Condens. Matter* **27**, 176002 (2015).
- 14 Freitas, D. C. Antiferromagnetism and ferromagnetism in layered 1T-CrSe₂ with V and Ti replacements. *Phys. Rev. B* **87**, 014420 (2013).

Reviewers' Comments:

Reviewer #2:

Remarks to the Author:

Authors answered my additional question appropriately. I believe that the manuscript is now ready for the publication.

Reviewer #3:

Remarks to the Author:

Dear Authors,

I appreciate the Authors for the appropriate revisions.